# Variation in Verification: Understanding Verification Dynamics in Large Language Models

**Yefan Zhou**[1,2]*, **Austin Xu**[1]†, **Yilun Zhou**[1]†, **Janvijay Singh**[1,3]*, **Jiang Gui**[2], **Shafiq Joty**[1]

[1]Salesforce AI Research    [2]Dartmouth College    [3]University of Illinois Urbana-Champaign

## Abstract

Recent advances have shown that scaling test-time computation enables large language models (LLMs) to solve increasingly complex problems across diverse domains. One effective paradigm for test-time scaling (TTS) involves LLM generators producing multiple solution candidates, with LLM verifiers assessing the correctness of these candidates without reference answers. In this paper, we study generative verifiers, which perform verification by generating chain-of-thought (CoT) reasoning followed by a binary verdict. We systematically analyze verification dynamics across three dimensions – problem difficulty, generator capability, and verifier generation capability – through empirical studies on 12 benchmarks across mathematical reasoning, knowledge, and natural language reasoning tasks using 14 open-source models (2B to 72B parameter range) and GPT-4o. Our experiments reveal three key findings about verification effectiveness: (1) Easy problems allow verifiers to more reliably certify correct responses; (2) Weak generators produce errors that are easier to detect than strong generators; (3) Verification ability is generally correlated with the verifier's own problem-solving capability, but this relationship varies with problem difficulty. These findings reveal opportunities for optimizing basic verification strategies in TTS applications. First, given the same verifier, some weak generators can nearly match stronger ones in post-verification TTS performance (e.g., the Gemma2-9B to Gemma2-27B performance gap shrinks by 75.7%). Second, we identify cases where strong verifiers offer limited advantages over weak ones, as both fail to provide meaningful verification gains, suggesting that verifier scaling alone cannot overcome fundamental verification challenges.

 Code    Project

## 1 Introduction

Large language models (LLMs) have advanced rapidly in solving reasoning tasks such as mathematics and code generation, yet their outputs remain unreliable, often containing subtle or obvious mistakes (Ke et al., 2025; Lightman et al., 2023). LLM based verification (Angelopoulos et al., 2025; Huang et al., 2024; Mao et al., 2024; Pan et al., 2025; Pandit et al., 2025) has emerged as a central mechanism to identify such errors in a scalable manner. Recent work has increasingly focused on *generative verifiers* (Liu et al., 2025d; Mahan et al., 2024; Zhang et al., 2025), which frame verification as next-token prediction: the model typically generates a chain-of-thought (CoT) reasoning trace and then outputs a binary verdict token. This approach has been shown to outperform earlier discriminative verifiers or scalar reward models (RMs, Lightman et al., 2023), as it better leverages the inherent text-generation capabilities of LLMs. One valuable downstream application of automatic verification is test-time scaling (TTS), where additional inference-time compute is allocated to improve generation performance. A popular paradigm of TTS is the use of a verifier model to evaluate candidate responses, filter errors, and identify correct solutions. This approach underlies techniques such as rejection sampling (Brown et al., 2024), re-ranking (Zhou et al., 2025), weighted majority voting (Wang et al., 2024a; 2023), and step-level generation (Snell et al., 2025).

Current practice in LLM verification often deploys strong, typically closed-source frontier models as verifiers. This practice rests on the assumption that verification quality scales with a verifier's capability to solve the same problem (i.e., its generation capability), a correlation demonstrated in recent

---

*Work done during an internship at Salesforce AI Research
†Work done at Salesforce AI Research

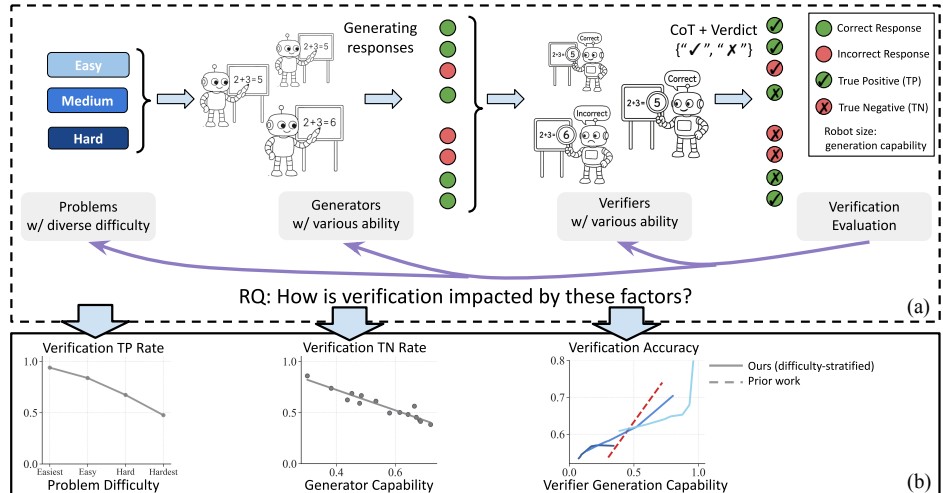

Figure 1: **Overview of our study on verification dynamics.** (a) We consider generative verification: an LLM generator produces a solution to a problem, and an LLM verifier conditions on the problem and solution to generate a verification CoT followed by a binary verdict ("Correct"/"Incorrect"). We design controlled experiments that vary problem difficulty, generator generation capability, and verifier generation capability, investigating how each of these factors influences verification performance. (b) Our analysis reveals three patterns: problem difficulty governs recognition of correct responses (true positives); generator generation capability determines error detectability (true negatives); and verifier generation capability correlates with performance in a difficulty-dependent manner, revealing non-linear regimes left uncovered in prior work. The three plots were generated by aggregating benchmark data across three domains and averaging performance metrics over 15 models.

work (Chen et al., 2025c; Krumdick et al., 2025; Tan et al., 2025). However, this practice may be sub-optimal given that verifying a solution is often easier than generating one from scratch, a phenomenon referred to as "verification asymmetry" (Wei, 2025). This asymmetry appears in several fields. In convex optimization, dual certificates enable efficient validation of optimality of a proposed solution, while in factorization, verifying correctness is trivial compared to finding the prime factors. Thus, it is worth investigating verification as a distinct capability rather than merely a byproduct of generation.

Despite extensive research on generation dynamics and the factors influencing generation quality (Allen-Zhu & Li, 2025; Chen et al., 2024; Ye et al., 2025), the dynamics of verification remain largely unexplored. In particular, little is known about how problem characteristics, properties of generated responses, and model capabilities interact to determine verification effectiveness. Without understanding verification dynamics, one can risk misallocating computational resources by defaulting to expensive frontier models when simpler alternatives might suffice. This gap in understanding motivates our central research question: *what factors influence verification success?*

In this paper, we present a systematic study of generative verification across three dimensions–problem difficulty, generator capability, and verifier generation capability–shown in Figure 1. We quantify verification performance by measuring the probability of the verifier recognizing both correct and incorrect generated solutions in controlled experimental settings. We focus on verifiable problems with objective ground-truth answers in mathematical reasoning, knowledge question-answering (QA), and natural language (NL) reasoning domains. This allows us to objectively measure verifier and generator performance, while simulating the reference-free evaluation settings where verifiers are typically deployed in practice, e.g., in TTS. While our experiments use these domains as a testbed, we believe the insights should extend to any domain where correctness can be reliably defined and checked.

**Main Findings.** While prior work showed that verifier generation capability correlates with verification performance (Chen et al., 2025c; Krumdick et al., 2025; Tan et al., 2025), we reveal that two additional factors, *problem difficulty* and *generator capability*, also critically influence verification success, as illustrated in Figure 1. Our analysis reveals:

- Problem difficulty primarily governs the recognition of correct solutions: verifiers are more likely to recognize correct solutions on easy problems than on difficult ones.

- Generator capability influences error detection: errors made by weak generators are easier to detect than those made by strong generators.
- Verifier generation capability correlates with verification performance in a manner dependent on problem difficulty: saturated (or uncorrelated) for easy problems, linear for medium problems, and threshold-limited for hard problems.

Our empirical analysis includes 2,347 math problems from eight datasets, 1,196 knowledge QA problems, and 901 NL reasoning problems, evaluated across 14 open-source models and GPT-4o.

**Application to TTS.** We demonstrate the practical implications of our findings for TTS. First, given the same verifier, the TTS performance of a weak generator can nearly match the performance of a strong generator. For instance, Gemma2-9B achieves comparable performance to Gemma2-27B when both use the same verifier, GPT-4o. Second, we identify regimes where a strong verifier (e.g., GPT-4o) offers no additional benefit and can be replaced by a weaker verifier (e.g., Qwen2.5-7B), with both providing limited gains. This occurs with strong generators or with problems at either extreme of the difficulty spectrum.

## 2 RELATED WORK

**Automatic Evaluation.** The deployment of LLMs as evaluators has emerged as a central mechanism for scalable assessment, with efforts focusing on training specialized small evaluators through fine-tuning (Singh et al., 2026; Wang et al., 2025; Whitehouse et al., 2025; Xu et al., 2025; 2026; Zhang et al., 2025). Beyond reference-based verifiers (Chen et al., 2025a; Liu et al., 2025b), verification approaches include self-verification (Chen et al., 2023; Huang et al., 2023; 2024; Kumar et al., 2024; Shinn et al., 2023), where models reflect on or critique their own outputs, and multi-agent verification (Li et al., 2023; Lifshitz et al., 2025; Zhuge et al., 2024), where multiple agents collaborate in debate-style or hierarchical setups to improve reliability. Prior work identifies several factors influencing evaluation performance, with evaluator generation capability being particularly important. Krumdick et al. (2025) find that evaluator performance changes significantly based on whether the evaluator is capable of answering the question or not. Tan et al. (2025) demonstrate the correlation between pairwise judging ability and generation ability on the same set of problems. Chen et al. (2025b) observe linear relationships between evaluation improvements and reasoning-required sample proportions in fine-tuned evaluators. Chen et al. (2025c) show a strong positive correlation between generation capability and evaluation accuracy. Our work extends these findings by identifying unexplored factors and showing that the relationship between evaluator generation capability and evaluation quality is more nuanced than previously understood. Lu et al. (2025) is a concurrent work with a similar research objective, finding that cross-family verification is particularly effective, complementing our analysis of how problem difficulty and model capability shape verification dynamics.

**Verification for Test-Time Scaling.** Early studies explore how to effectively apply verification methods to improve TTS performance. Snell et al. (2025) show RMs improve various TTS approaches, including Best-of-N and beam search, while Liu et al. (2025a) find that compute-optimal strategies vary with policy models and problem difficulty. Recent work explores alternatives to discriminative RMs: Zhang et al. (2025) show trained generative verifiers outperform RMs in Best-of-N, and Zhou et al. (2025)'s JETTS benchmark demonstrates generative evaluators match outcome RMs in reranking. While verification benefits from increased model size and test-time compute, recent work addresses how to reduce these computational costs. Saad-Falcon et al. (2025) propose a framework to aggregate weak verifiers to approach strong ones; Angelopoulos et al. (2025) balance weak/strong evaluators for efficiency; Stroebl et al. (2024) analyze fundamental limits of resampling with imperfect verifiers; and Singhi et al. (2025) propose strategies to balance solving-verification trade-offs. Our work studies the factors driving verification and explores their implications for TTS.

## 3 EXPERIMENTAL SETUP

### 3.1 PRELIMINARIES

**Problem and Response Space.** Let $x$ denote a problem with ground-truth answer $y^*(x)$. A model response $r$ to $x$ consists of a CoT solution and a final answer $a(r)$, and we consider the response correct if $a(r) = y^*(x)$. As discussed in Section 1, our study uses verifiable problems with objective answers, allowing us to rigorously evaluate verifier outputs against ground-truth while simulating reference-free evaluation settings.

**Generator and Verifier.** A generator $G$ maps a problem $x$ to a distribution over responses, denoted $r \sim G(\cdot|x)$. A verifier $V$ takes a problem–response pair $(x, r)$ and outputs a judgment of correctness.

In the binary case, $V(x, r) \in \{0, 1\}$, where 1 indicates acceptance and 0 indicates rejection. More generally, a generative verifier produces a verification CoT explaining its reasoning, followed by an explicit verdict such as "Correct" or "Incorrect." The prompt templates are provided in Appendix A.

**Generation Capability.** We measure the generation capability of a model using its *pass rate*. For a generator $G$ and problem $x$, we define $p_G(x) = \Pr[a(r) = y^*(x) \mid r \sim G(\cdot|x)]$ as the pass rate on a single problem, i.e., the probability that $G$ solves $x$ correctly on one sampled attempt. We define $p_G(\mathcal{D}) = \frac{1}{|\mathcal{D}|} \sum_{x \in \mathcal{D}} p_G(x)$ as the pass rate aggregated over a dataset $\mathcal{D}$, which we use as the overall measure of a model's generation capability. Since the verifiers we study are generic LLMs (e.g., GPT-4o), we measure a verifier's generation capability using the same metric by prompting it as a generator.

We estimate these pass rates empirically as $\hat{p}_G(x; K)$ and $\hat{p}_G(\mathcal{D}; K)$ by sampling $K$ responses per model-problem pair. Since $K$ is fixed at 64 throughout, we often omit $K$ for simplicity, writing $\hat{p}_G(x)$ and $\hat{p}_G(\mathcal{D})$. We use temperature 0.7 and top-p 1.0 as default sampling hyperparameters, and adopt recommended settings when available (e.g., temperature 0.7 and top-p 0.8 for the non-thinking mode of Qwen3). Ground-truth correctness is established with `Math-Verify` (Kydlíček, 2025), supplemented by LLM-as-a-judge grading to reduce false negatives (details in Appendix B.1). These 64 responses per problem–model pair are used to estimate generation capability and problem difficulty.

**Problem Difficulty.** We define the difficulty of a problem as the average pass rate across a set of diverse generators $\mathcal{G}$, $d(x) = \frac{1}{|\mathcal{G}|} \sum_{G \in \mathcal{G}} \hat{p}_G(x)$. This score reflects how broadly solvable a problem is: if most generators succeed, $d(x)$ is high (easy problem), while if few succeed, $d(x)$ is low (hard problem). It provides a model-agnostic way to partition problems by difficulty, extending prior work (Snell et al., 2025), which measured difficulty relative to a single generator.

**Verification Metrics and Evaluation.** We evaluate verifiers using true positive rate (TPR), the probability of the verifier accepting a correct response: $\text{TPR} = \mathbb{E}[V(x, r) \mid a(r) = y^*(x)]$, and true negative rate (TNR), the probability of rejecting an incorrect response: $\text{TNR} = \mathbb{E}[1 - V(x, r) \mid a(r) \neq y^*(x)]$.[1] We also report balanced accuracy, $\text{Acc}_{\text{bal}} = \frac{1}{2}(\text{TPR} + \text{TNR})$, which accounts for class imbalance. For verification evaluation, we subsample 8 responses from each 64-sample pool, balanced with 4 correct and 4 incorrect when possible. For very hard problems with fewer than 4 correct responses, we keep all correct ones and sample incorrect ones to reach 8 total (and vice versa for easy problems). Each verifier evaluates responses from all 15 models over the full test set using greedy decoding, unless a controlled subset is specified.

**Verification-Augmented Test-time Scaling.** We consider the TTS setting of sampling multiple responses from the generator and filtering with a verifier before evaluation. For each problem $x \in \mathcal{D}$, we sample $K$ responses from the generator using a fixed temperature, with $K = 64$ in our experiments. Without verification, TTS performance is measured as $\hat{p}_G(\mathcal{D}; K)$ (or $\hat{p}_G(\mathcal{D})$), the empirical pass rate defined above. With verification, the verifier $V$ evaluates each candidate, and only responses deemed "Correct" are retained for evaluation. The performance of verification-augmented TTS is measured as

$$\hat{p}_{G,V}(\mathcal{D}; K) = \frac{1}{|\mathcal{D}|} \sum_{x \in \mathcal{D}} \left( \frac{1}{K'} \sum_{i=1}^{K} \mathbb{1}\big(a(r_i) = y^\star(x)\big) \cdot V(x, r_i) \right) \tag{1}$$

where $K' = \sum_{i=1}^{K} V(x, r_i)$. This metric represents the conditional pass rate, i.e., the fraction of correct responses among those retained by the verifier. A corner case arises when the verifier rejects all responses ($K' = 0$); in this case, we set the metric to the generator's pass rate $\hat{p}_G(\mathcal{D})$, so evaluation reverts to selecting from the original $K$ responses in the non-verified setting. We define the *verification gain* from verifier $V$ as the difference relative to the performance without verification, $\Delta \hat{p}_V = \hat{p}_{G,V}(\mathcal{D}) - \hat{p}_G(\mathcal{D})$, which quantifies how much gain can be attributed to verification. Note that our formulation of TTS differs from the common setting where a single "best" response (e.g., by majority vote) is selected and then evaluated. Instead, we report the empirical pass rate of the verifier-retained pool, which can be interpreted as the expected accuracy of uniformly sampling one response from that pool. This expectation-based view captures the average quality of verifier-retained responses without tying performance to a specific selection strategy.

---

[1]If the verifier generates an invalid output (e.g., due to the CoT running out of max generation length), we treat it as an uninformative verdict of "Correct" and "Incorrect" each with probability of $50\%$. Computationally, we set $V(x, r) = 0.5$ in this case, and also in Equation 1.

## 3.2 TASKS AND MODELS

**Mathematical Reasoning.** We collect a total of 2,347 problems from the test sets of eight mathematical reasoning benchmarks: GSM8K (Cobbe et al., 2021), MATH500 (Hendrycks et al., 2021), OlympiadBench (He et al., 2024), AIME24/25 (Li et al., 2024), AMC23 (Li et al., 2024), Minerva-Math (Lewkowycz et al., 2022), and BBEH Multi-step Arithmetic (Kazemi et al., 2025). We use the entire test sets of these benchmarks, except for GSM8K, from which we subsample 600 of 1,319 problems to balance difficulty distribution and reduce the proportion of easy problems.

**Knowledge.** We use a subset of MMLU-Pro (Wang et al., 2024b) as our knowledge category. We randomly subsample 10% from each of its 14 disciplines, yielding 1,196 problems. MMLU-Pro consists of college-level multiple-choice questions spanning STEM, humanities, and social sciences.

**Natural Language Reasoning.** We collect 901 multiple-choice problems from three benchmarks. (1) ReClor (validation set, Yu et al., 2020), a multiple-choice benchmark requiring logical analysis of short passages. (2) FOLIO (Han et al., 2022), a first-order logic reasoning benchmark in natural language. (3) GPQA Diamond (Rein et al., 2024), a dataset that consists of graduate-level multiple-choice science questions, requiring multi-step reasoning.

**Models.** We use 14 open-source models from four families: (1) Qwen2.5 at 3B, 7B, and 72B (Team, 2024); Qwen3 at 4B, 8B, and 32B (Yang et al., 2025); (2) Llama-3.2 at 3B, Llama-3.1 at 8B, and Llama-3.3 at 70B (Grattafiori et al., 2024); (3) Gemma-2 at 2B, 9B, and 27B (Team et al., 2024); (4) Ministral 8B and Mistral-Small-24B; and one closed-source model GPT-4o (Hurst et al., 2024). All models are instruction-tuned versions by default. Each model is used both as a generator and a verifier. We use abbreviated model names in figures for space efficiency; see Appendix B.2 for mappings.

## 4 EXPERIMENTAL RESULTS

Our experiments focus on how problem difficulty and generator and verifier generation capability influence verification performance. We present the three research questions and main findings below.

- **RQ1: How does problem difficulty affect verification?** (Section 4.1) TPR increases steadily with decreasing problem difficulty, meaning verifiers better recognize correct responses on easier problems. However, TNR shows no predictable relationship with problem difficulty. This indicates that problem difficulty primarily influences correctness recognition.
- **RQ2: How does the generator's generation capability influence verification?** (Section 4.2) As generators become stronger, TNR decreases substantially while TPR increases only slightly. This reveals that generator capability primarily determines error detectability: stronger generators produce errors that are harder for verifiers to identify.
- **RQ3: How does verifier generation capability impact verification?** (Section 4.3) Verifier generation capability and verification performance are generally positively correlated. However, the form of correlation depends heavily on problem difficulty: linear correlation occurs in medium-difficulty problems, while nonlinear patterns appear in other difficulty levels.

Throughout this section, we use the oracle problem difficulty measure defined in Section 3.1, which relies on ground-truth labels, to avoid measurement error. In Appendix D.1, we validate all key findings using a practical, label-free difficulty estimator and show that the trends remain consistent.

### 4.1 HOW DOES PROBLEM DIFFICULTY AFFECT VERIFICATION?

To examine how problem difficulty influences verification, we partition problems into four equal-sized quartiles by their difficulty score $d(x)$, termed "hardest", "hard", "easy", and "easiest".

**Problem difficulty primarily influences the verifier's ability to recognize correct responses.** Our analysis is conducted at two levels of granularity: response level and problem level. Both analyses reveal that problem difficulty mainly shapes the verifier's sensitivity to correct responses, while not consistently affecting its ability to identify incorrect responses.

At the response level, we compute the TPR and TNR of all responses within each difficulty quartile. As shown in Figure 2, TPR increases steadily as problems become easier, while TNR shows no clear trend. This pattern is consistent across model families and domains. At the problem level, we pool responses from all generators for each problem and compute a single TPR and TNR per problem. The distribution of these metrics within each quartile is reported in Figures 9 and 10 of Appendix C.1. We observe that easier problems yield higher and more stable TPR, while harder problems exhibit lower and more variable TPR. In contrast, TNR distributions show no consistent

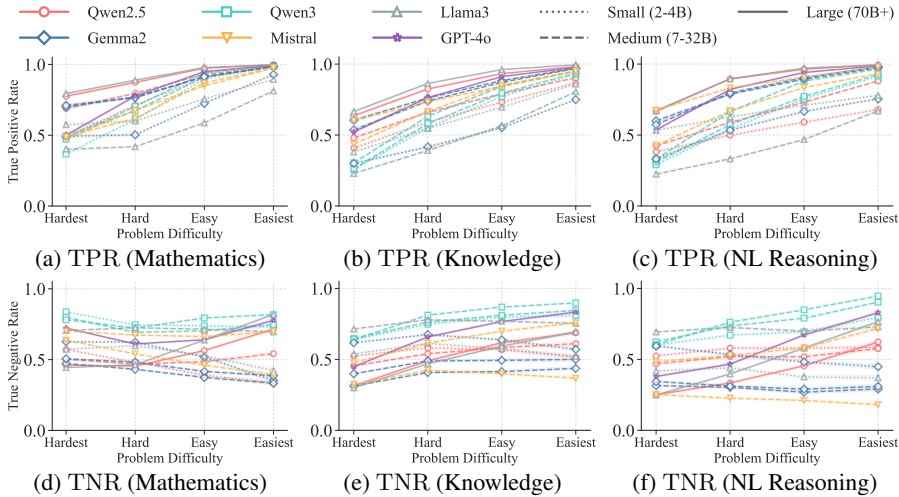

Figure 2: **Problem difficulty primarily affects** TPR **of verification.** Each curve shows verifier performance across four difficulty groups, with the $x$-axis indicating problem difficulty and the $y$-axis reporting TPR (a-c) and TNR (d-f). Colors denote model families, and line styles indicate model size.

correlation with problem difficulty. In Appendix C.2, we show that the main verification dynamics about TPR we identified generalize to reasoning models, while extended reasoning provides benefits and alters TNR behavior. We additionally confirm that this trend persist for a large open-source model (Qwen3-235B) in Appendix D.3 and across different verification prompts in Appendix D.4.

To understand this pattern, case studies in Figure 32 show that verifiers tend to generate their own reference solutions for comparison during verification. As the problem difficulty increases, these verifier-generated answers become increasingly incorrect, producing false negatives (FNs) that reduce TPR. We support this explanation with a large-scale analysis. We use LLM-as-judge to detect whether a verification CoT contains solving mistakes, with experimental details provided in Appendix B.3. As we show in Figure 3, as problem difficulty increases, the verifier increasingly makes mistakes in generating the reference answer, with 39.1% of verification FNs containing a reference answer mistake in the hard set of problems. This shows that faulty reference generation is a large driving factor of FNs.

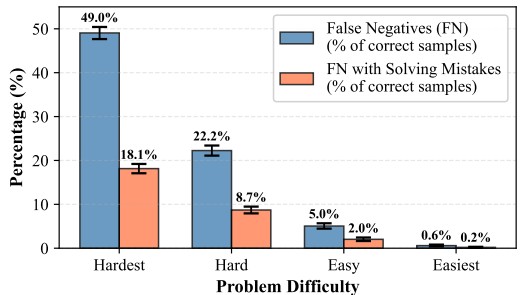

Figure 3: **Interpreting RQ1 finding by analyzing solving mistakes in false-negative verifications.** Blue: percentage of FNs among sampled correct responses. Orange: percentage of those FNs whose verification CoTs contain solving mistakes. Error bars: 95% CIs.

## 4.2 HOW DOES GENERATOR CAPABILITY INFLUENCE VERIFICATION?

We study how generator capability affects verifier performance by having each verifier evaluate responses from each generator. Generators of different capabilities may produce extreme response distributions, e.g., weak generators may produce no correct response on hard problems within 64 samples. To ensure fair comparison, we compute TPR on problem subsets where all generators produce at least one correct response. Analogously, TNR is computed on problems where all generators produce at least one incorrect response. Details are provided in Appendix B.4.

As shown in Figures 4a to 4c, TPR remains uniformly high across nearly all settings and increases further with stronger generators. The heatmap is dominated by red colors, with values mostly above 0.7, indicating that most verifiers are already reliable at recognizing correct responses. As generator capability improves, TPR approaches 1.0. This suggests that generator strength influences recognition of correct responses in a relatively mild way.

**Generator capability correlates with error detection in verification.** In Figures 4d to 4f, moving from weaker generators on the left to stronger ones on the right, the heatmap shifts generally from

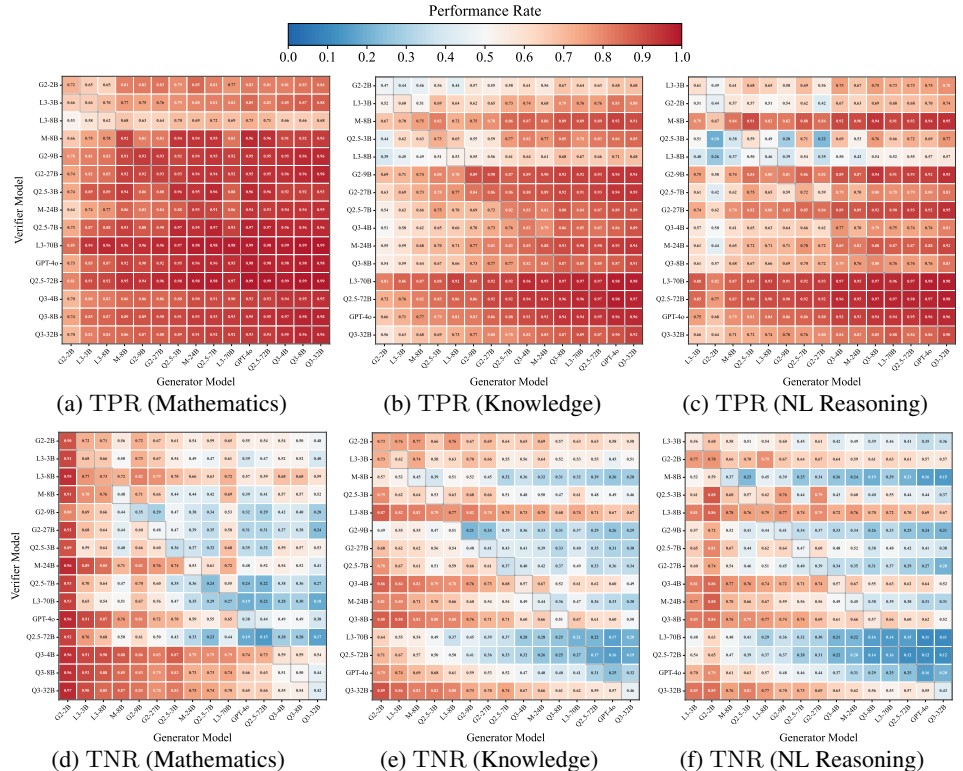

Figure 4: **Generator capability influences verifier performance of identifying incorrect responses.**
Heatmaps show (a-c) TPR and (d-f) TNR when pairing 15 verifier models (rows) with 15 generator
models (columns). Rows and columns are ordered by models' generation capability computed on
all problems of each domain. Values indicate mean performance over the evaluation subset.

red to blue, indicating a substantial decrease in TNR. For example, in the Mathematics domain, for
the Qwen2.5-72B verifier, TNR drops from 0.68 on solutions generated by Llama-3.1-8B to 0.17 on
those by Qwen3-32B. The overall pattern is consistent across three domains and nearly all verifiers.
These results show that generator capability strongly modulates the detection of incorrect responses.
In Appendix C.2, we show that this finding generalizes to reasoning models. We further verify that
this trend remains robust when scaling to a large open-source model (Qwen3-235B; Appendix D.3)
and when varying the verification prompt (Appendix D.4).

We interpret this phenomenon through case studies in Figure 33. Strong generators produce internally
consistent reasoning chains where early mistakes (e.g., missed cases) propagate coherently, yielding
well-structured but incorrect solutions that cause the verifier false positives. Weak generators
produce surface-level errors such as self-contradictions, facilitating verifier rejection. We support this
explanation with a large-scale analysis. We use an independent LLM-as-judge to determine whether
a generator's CoT contains surface-level errors, with experimental details provided in Appendix B.3.
The results in Figure 5 show a clear trend: as generator capability increases, the frequency of
surface-level errors decreases. This is consistent with Yin et al. (2025), who find that error patterns
of stronger models shift from basic computational mistakes to more sophisticated reasoning failures.

### 4.3 HOW DOES VERIFIER GENERATION CAPABILITY IMPACT VERIFICATION?

We measure verifier generation capability and evaluate verification performance using balanced
accuracy ($\text{Acc}_{\text{bal}}$) on the entire test set. Each verifier is evaluated on responses from all generators,
and we report results both averaged across all problems and stratified by problem difficulty. To
characterize the relationship between generation capability and verification performance, we employ
locally weighted regression (Cleveland, 1979) with a bandwidth of 0.6 to fit nonparametric curves.
We compare $R^2$ values between nonparametric and linear fits to assess linearity. We also report the
Pearson correlation coefficient (Benesty et al., 2009) as another measure of linear correlation.

Figures 6a to 6c show a strong overall correlation between verifier generation capability and verifica-
tion accuracy, with NL reasoning showing less linearity than other domains. This result is consistent

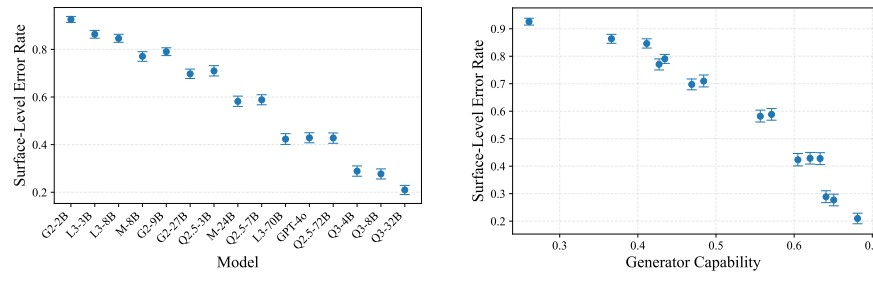

(a) Surface-level Error Rate vs. Name

(b) Surface-level Error Rate vs. Capability

Figure 5: **Interpreting RQ2 finding by analyzing surface-level errors in generators' solutions.**
(a) The $x$-axis lists generator models sorted by generation capability; the $y$-axis shows the percentage
of responses containing surface-level errors. (b) The $x$-axis shows the capability values of the same
generators; the $y$-axis is the same as in (a). Error bars indicate 95% CIs.

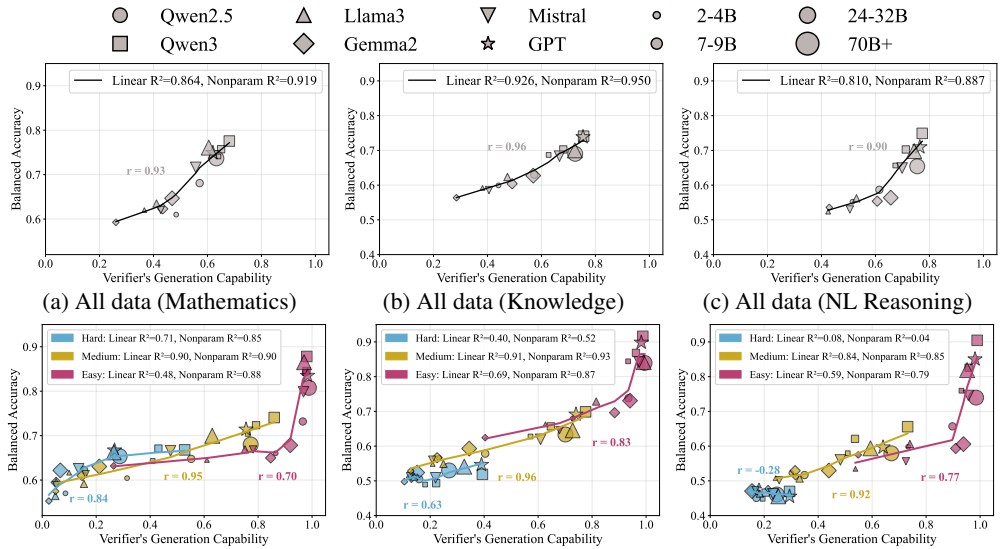

Figure 6: **Distinct correlation forms between verification performance and generation capability.**
Solid lines represent nonparametric fits to the data; $r$ indicates the Pearson correlation coefficient. (a-
c) Averaged across all problems, verifier generation capability exhibits a strong linear correlation with
balanced accuracy. (d-f) When stratified by problem difficulty, distinct correlation patterns emerge.

with prior work showing that evaluator accuracy tends to track the evaluator's task performance, with
the relationship appearing nearly linear. While this global trend validates findings in prior work (Chen
et al., 2025c; Tan et al., 2025), a closer inspection of the trend reveals highly non-linear regimes.

**Verifier generation capability influences verification accuracy differently based on problem
difficulty.** Stratified analysis reveals regime-dependent correlation with phase-transition behavior.
We partition problems into 10 equal-width bins by difficulty $d(x)$ and analyze three representative
intervals: *hard* $[0.1, 0.3]$, *medium* $[0.4, 0.5]$, and *easy* $[0.8, 0.9]$ in Figures 6e, 6f and 23a. For hard
problems (blue), verification accuracy shows minimal improvement with increasing capability.
Mathematics plateaus around 0.65 accuracy after initial gains, while other domains remain flat
throughout. Notably, verifiers achieve below-random accuracy on hard NL Reasoning problems,
which we analyze in Appendix C.3. Medium problems (yellow) exhibit steady accuracy increases
with capability, indicating strong linear relationships. This is confirmed by linear and nonparametric
fits that yield nearly identical $R^2$ values, with $r > 0.9$. Easy problems exhibit a threshold effect
at the $x$-axis around 0.9: below this threshold, the relationship is linear; above it, small capability
improvements yield large verification gains. Hard and easy regimes show nonlinearity with
nonparametric $R^2$ exceeding linear $R^2$ by 0.1–0.2 and $r < 0.85$. The exception is NL Reasoning
on hard problems, where both fits yield near-zero $R^2$, indicating no meaningful capability-accuracy

relationship. Appendix C.4 provides additional results (Figure 14), including analysis of why different patterns are observed in varying difficulty intervals.

## 5 APPLICATION TO TEST-TIME SCALING (TTS)

Our analysis in Section 4 is conducted with verification itself as the end goal. However, our findings have direct implications for TTS. We analyze two research questions in TTS settings that naturally arise out of our previous findings, and present our results below:

- **RQ4: Given a fixed verifier, can a weak generator match a stronger generator in TTS?** (Section 5.1) Weak generators can nearly match stronger generators' post-verification performance. Verification gains peak at weak-medium generators by achieving a high error detection rate (TNR) while maintaining a moderately high correctness recognition rate (TPR).
- **RQ5: Can weak verifiers match the gains of strong verifiers in TTS?** (Section 5.2) The verification gain gap between weak and strong verifiers narrows at both low and high problem difficulty extremes, and when using strong generators.

The following sections present results on the Mathematics domain, with complete results across all three domains in Appendices C.5 and C.6. In Appendix D.2, we show that these findings can guide generator and verifier selection in TTS and lead to efficiency improvements.

### 5.1 CAN WEAK GENERATORS MATCH STRONGER GENERATORS IN TTS?

We evaluate TTS with a fixed verifier (GPT-4o) by varying generator capability and reporting pass rates before and after verification, along with the verification gain $\Delta \hat{p}_V$.

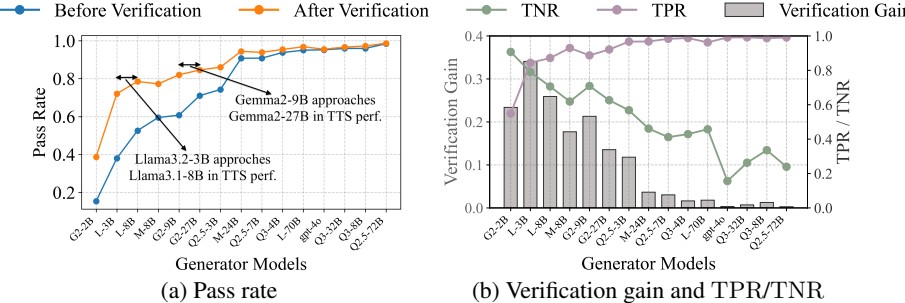

(a) Pass rate          (b) Verification gain and TPR/TNR

Figure 7: **TTS performance before and after verification when varying generator strength.** Results are reported on problems with difficulty in the range [0.7, 0.8) from the Mathematics domain, including 181 problems. (a) Pass rate before (blue) and after (orange) adding a fixed verifier (GPT-4o). The generators in the $x$-axis are ordered from weaker (left) to stronger (right) by generation capability measured on the problem subset. (b) Bar chart shows the verification gain $\Delta \hat{p}_V$ (left $y$-axis) for each generator. Lines show the verifier's TNR and TPR (right $y$-axis).

**Verification gain peaks for weak–medium generators, enabling them to approach stronger models post-verification.** As shown in Figure 7a, weak generators start with much lower pass rates but improve dramatically after verification, reaching levels comparable to larger models. For example, Gemma2-9B starts from a significantly lower baseline but, after verification, achieves a pass rate nearly matching Gemma2-27B. The performance gap shrinks from 10.3% to 2.5%, closing 75.7% of the original difference. Figure 7b explains this phenomenon: as generator strength increases (left to right), TNR decreases sharply while TPR rises only modestly, consistent with **RQ2** findings. Consequently, verification gain (gray bars) peaks at weak-medium generators, which achieve high TNR for effective error filtering while maintaining moderate TPR to preserve correct responses. For the strongest generators, errors become harder to identify, causing TNR decline and limiting gains. In Appendix C.5, we show the findings derived from the Mathematics domain generalize well to two other domains in Figure 15. We provide additional evidence confirming the generalizability: first, verification gains peak for weak-medium generators across a broad range of problem difficulties ($d(x) \geq 0.3$) in all domains (Figures 16 to 18); second, performance gaps of most weak and strong model pairs can be reduced by verification when evaluated on the entire domain datasets (Figure 19), mostly achieving 30-50% reduction. These results suggest that weak generators, when paired with a strong verifier, can serve as a cost-effective alternative to strong generators. One might question, however, whether such a pairing is truly efficient given the cost of a strong verifier (GPT-4o). In Appendix D.5, we address this with a case study showing that pairing a weak generator with a strong verifier can match the performance of a strong generator alone while reducing token consumption.

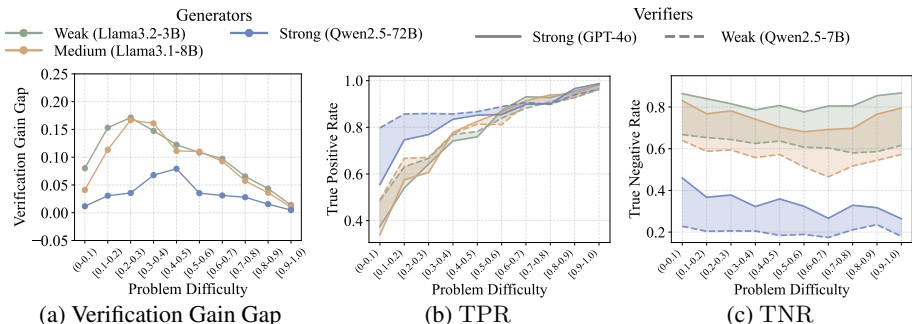

(a) Verification Gain Gap      (b) TPR      (c) TNR

Figure 8: **Weak vs. strong verifiers under varying problem difficulty and generator strength.** The $x$-axis shows problem difficulty, ordered from hardest to easiest, measured relative to each generator. (a) Verification gain gap between strong and weak verifiers when applied to weak, medium, and strong generators. (b) TPR increases as problems become easier; shadow band indicates TPR gap between two verifiers. (c) As generators strengthen, TNR decreases overall and the TNR gap (shadow band) narrows. Results are from the Mathematics domain.

## 5.2 CAN WEAK VERIFIERS MATCH THE GAINS OF STRONG VERIFIERS IN TTS?

We analyze the verification gain gap between a strong verifier (GPT-4o) and a weaker one (Qwen2.5-7B) across problem difficulty ranges and generator strengths. The verification gain $\Delta\hat{p}_V$ is defined in Section 3 and the gap between verifier is $\Delta\hat{p}_{V_{\text{strong}}} - \Delta\hat{p}_{V_{\text{weak}}}$. Our goal is to identify when this gap narrows, as such regimes suggest weak verifiers can substitute for strong ones.

**The gap narrows on the extremes of problem difficulty** As shown in Figure 8a, the verification gain gap shrinks as problems become easier, which corresponds to the rising TPR for both weak and strong verifiers seen in Figure 8b. This aligns with our **RQ1** findings that easier problems improve TPR for all verifiers. Even weak verifiers reliably recognize correct responses on easy problems, leaving little room for strong ones to provide additional benefit. At the opposite extreme, the gap also narrows on the hardest problems. As discussed in **RQ3** and shown in Figures 21d to 21f, increasing verifier generation capability (or scaling up to larger models) fails to improve verification accuracy on hard problems, resulting in only marginal performance differences between weak and strong verifiers.

**The gap narrows as generators become stronger.** Figure 8 shows that increasing generator capability reduces the difference between weak and strong verifiers. This is consistent with **RQ2** (Section 4.2), where we observed that the verifier's TNR decreases as the generator capability increases. As both weak and strong verifiers experience lower TNR, the gap between them also shrinks, shown as the narrowing shaded band between solid and dashed curves in Figure 8c.

The results here are obtained from the Mathematics domain, and we show the findings generalize to two other domains in Appendix C.6. In regimes of very easy/hard problems or when evaluating strong generators' responses, weak verifiers provide gains to TTS performance comparable to strong verifiers. However, these convergence regimes coincide with minimal verification benefit overall. Figures 21a to 21c shows verification gains drop to 0.1 or below for both verifiers on easy and hard problems, verification on strong generators yields peak gains of only 0.1, precisely where the gap narrows. Thus, while weak and strong verifiers converge in these regimes, this convergence occurs where both provide minimal practical value. This reveals that scaling verifiers from 7B models to GPT-4o fails to overcome fundamental verification challenges, with GPT-4o providing limited improvement over small open-source models in the identified regimes.

## 6 CONCLUSION

We study LLM verification across problem difficulty, generator capability, and verifier generation capability, revealing that verification success depends on their interactions. We find that problem difficulty primarily shapes correct solution recognition, generator capability influences error detectability, and verifier generation capability correlates with verification in problem difficulty-dependent patterns. We examine the implications of these findings for verification deployment in TTS, identifying both opportunities and limitations. Stronger generators may not be necessary, as weaker generators can approach the post-verification performance of stronger ones when paired with a fixed verifier. This suggests potential for strategic model pairing that could reduce computational costs in verifier-based TTS methods. Our results also identify regimes where investing in larger verifiers yields no benefit, such as when evaluating responses from strong generators or problems at difficulty extremes.

ETHICS STATEMENT

This work exclusively evaluates large language models on publicly available academic benchmarks containing mathematical reasoning, knowledge, and natural language problems with objective ground-truth answers. All experiments involve automated evaluation of model outputs without human subject participation. The datasets used are established research benchmarks designed for educational problem-solving tasks. Our study aims to understand verification dynamics to improve the computational efficiency of LLM systems, posing no ethical concerns regarding privacy, harmful content generation, or potential misuse.

REPRODUCIBILITY STATEMENT

We provide comprehensive details to ensure reproducibility of our findings. All experiments use publicly available datasets and open-source/commercial LLMs. We specify the model names, versions, dataset sources, and inference hyperparameters in Section 3.2. Complete prompt templates for both generation and verification tasks are provided in Appendix A. The mathematical formulations of all metrics, along with estimation procedures and aggregation methods, are formally defined and clearly described in Section 3.1 and Appendix B.

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

APPENDIX

## THE USE OF LARGE LANGUAGE MODELS (LLMS)

LLMs were used exclusively as writing assistance tools in preparing this manuscript. Specifically, we employed LLMs for grammar checking. An LLM-based image generation tool was used to create the robot caricature in Figure 1. All research ideation, experimental design, analysis, and scientific conclusions are entirely the work of the authors. The LLMs played no role in the conception of research questions, methodology development, or interpretation of results. Authors take full responsibility for all content in this paper, including any text refined with LLM assistance.

## A  PROMPT TEMPLATES

**Response Generation Prompt.** Here, we provide the prompts to generate model responses to questions from three domains. For each model, we use its default system prompt as specified in the model documentation.

**Mathematical datasets**

```
### User Prompt
{problem}
Please reason step by step, and put your final answer within \\boxed{{}}.
```

**Knowledge: MMLU-Pro**

```
### User Prompt
The following are multiple choice questions (with answers) about {category}. Think step
by step and then output the answer in the format of \"The answer is (X)\" where X is the
correct letter choice.

Question: {question}
Options:
{options}

Answer: Let's think step by step.
```

**Natural Language Reasoning: ReClor**

```
### User Prompt
Answer the following multiple choice question. The last line of your response should be
of the following format: 'Answer: $LETTER' (without quotes) where $LETTER is one of
ABCD. Think step by step before answering.

{context}

{question}

{choices}

Output your answer strictly following this format:
Reasoning: <step-by-step reasoning>
Answer: <your choice>
```

**Natural Language Reasoning: FOLIO**

```
### User Prompt
{question}

Based on the above, can the following be inferred?
{conclusion}

Think step-by-step before outputting your answer, which must be "Yes", "No", or
"Uncertain"

Output your answer strictly following this format:
Reasoning: Your reasoning here
Answer: Yes or No or Uncertain
```

**Natural Language Reasoning: GPQA Diamond**

```
### User Prompt
Answer the following multiple choice question. The last line of your response should be
of the following format: 'Answer: $LETTER' (without quotes) where $LETTER is one of
ABCD. Think step by step before answering.

Question: {question}
Options:
{choices}

Output your answer strictly following this format:
Reasoning: <step-by-step reasoning>
Answer: <your choice>
```

**Verification Evaluation Prompt.** Below, we present the prompt template used to evaluate verification performance.

```
### System Prompt
Please act as an impartial judge and evaluate the correctness of the response provided
by an AI assistant to the user prompt displayed below. You will be given the assistant's
response.

When evaluating the assistant's response, identify any mistakes or inaccurate
information. Be as objective as possible. Avoid any biases, such as order of responses,
length, or stylistic elements like formatting.

Before providing an your final verdict, think through the judging process and output
your thoughts as an explanation

After providing your explanation, you must output only one of the following choices as
your final verdict with a label:

1. The response is correct: [[Correct]]
2. The response is incorrect: [[Incorrect]]

Use the following template:
Explanation: Your detailed thought process as an explanation.
Verdict: [[Correct]] or [[Incorrect]].

### User Prompt
<|User Prompt|>
{question}

<|The Start of Assistant's Answer|>
{response}
<|The End of Assistant's Answer|>
```

**Verification Evaluation Prompt (Solve-then-Verify).** Below, we present a variant of the prompt template used to evaluate verification performance. This template is adapted from the idea of Chen et al. (2025d); Sun et al. (2025) that prompts the model to solve the problem first and use its own solution as a reference for verification.

```
### System Prompt
Please act as an impartial judge and evaluate the correctness of the response provided
by an AI assistant to the user prompt displayed below. You will be given the assistant's
response.

First, you MUST solve the question yourself and put your final answer following the
format requested in <|User Prompt|>. Provide your own solution with final answer before
proceeding to the evaluation. When evaluating the candidate's response, you MUST refer
to your own solution.

Be as objective as possible. Avoid any biases, such as order of responses, length, or
stylistic elements like formatting.

After providing your explanation, you must output only one of the following choices as
your final verdict with a label:

1. The response is correct: [[Correct]]
2. The response is incorrect: [[Incorrect]]

Use the following template:
Solution: Your own reasoning and final answer to the problem.
Explanation: Your detailed thought process as an explanation.
Verdict: [[Correct]] or [[Incorrect]].

### User Prompt
<|User Prompt|>
{question}

<|The Start of Assistant's Answer|>
{response}
<|The End of Assistant's Answer|>
```

**Verification Evaluation Prompt (Step-by-Step).** Below, we adopt a variant of the verification template from Zhang et al. (2024). This template only has a user prompt.

```
### User Prompt
You are a math teacher. Grade the Solution, verifying correctness step by step.
At the end of the Solution verification, when you give your final grade, write it in the
form "Verification: Is the answer correct (Yes/No)? X", where X is either \\boxed{{Yes}}
or \\boxed{{No}}.
Question: {question}
Solution: {response}
```

**Fallback Correction Check Prompt.** When `Math-Verify` returns unparsable or incorrect results, we employ LLM-as-judge as a fallback mechanism for correctness verification. Below, we provide the prompt template used for this secondary verification step:

```
### User Prompt
Given a math problem, its correct answer, and the model's generated answer, determine if
the model's generated answer is correct.

VALIDATION CRITERIA:
1. Identify the final answer, which is usually put inside \\boxed{{answer}} or
**answer**.
2. The answer must be mathematically equivalent to the correct answer
3. The answer must be complete with a clear final result
4. The answer must not just contain similar numbers - it must reach the correct
conclusion
5. If the generated answer contains multiple different final answers or is ambiguous
about which is the final answer, mark it as 'False'

IMPORTANT: Just having the same numbers as the ground truth is NOT sufficient - the
model must actually solve the problem correctly and provide the correct final answer in
the designated format.

Respond with 'True' if the answer is correct and complete, and 'False' if it is
incorrect or incomplete.
Directly provide your judgement 'True' or 'False' without any other description.

Problem: {problem}
Correct Answer: {ground_truth_answer}
Model's Generated Answer: {model_response}
Your judgement:
```

**Verification Analysis Prompt for Detecting the Solving Behavior.** Below we present the prompt template used to analyze if verification CoTs contain solving behavior. This prompt asks the verifier to solve the problem first and then

```
### System Prompt
Please act as an impartial analyzer and determine whether the AI assistant, when
evaluating a candidate response, generates its own solution (full or partial) as a
reference - either explicitly or implicitly.

**Key Indicators that the assistant GENERATED its own solution [[Yes]]:**
- The assistant determines what the correct answer should be (even implicitly)
- The assistant works through the problem logic independently (e.g., "the premises
actually say X", "what can be inferred is Y")
- The assistant constructs its own interpretation of what follows from the problem
statement
- The assistant makes claims about what the "correct reasoning" would be

**Key Indicators that the assistant ONLY INSPECTED the candidate [[No]]:**
- The assistant only points out errors in the candidate's reasoning flow without
determining the correct answer
- The assistant only identifies missing steps or unsupported leaps in the candidate's
reasoning chain
- The assistant only verifies consistency within the candidate's own reasoning chain
- Focus is purely on "the candidate failed to justify X" rather than "X is actually Y"

**Important:** Even implicit or partial solutions count as generating a solution. If the
assistant reveals what it believes to be true/false about the problem, it has generated
a solution.

**Example 1:**
```

```
<|User Prompt|>
{example1_question}

<|The Start of Candidate Response|>
{example1_response}
<|The End of Candidate Response|>

<|The Start of Assistant's Evaluation|>
{example1_evaluation}
<|The End of Assistant's Evaluation|>

**Expected Analysis:**
{example1_analysis}

---

Now analyze the following case. Look carefully for ANY statement where the assistant
determines what is actually true, correct, valid, or inferable - these indicate the
assistant generated its own solution.

Use the following template:
Explanation: Your reasoning for why the assistant did or did not generate its own
solution. Quote specific phrases that reveal solution generation.
Verdict: [[Yes]] or [[No]]

### User Prompt
<|User Prompt|>
{question}

<|The Start of Candidate Response|>
{response}
<|The End of Candidate Response|>

<|The Start of Assistant's Evaluation|>
{evaluation}
<|The End of Assistant's Evaluation|>
```

**Verification Analysis Prompt for Detecting the Reasoning Mistakes.** Below, we present the
prompt template used to analyze if verification CoTs contain any basic reasoning mistakes.

```
### System Prompt
You are an impartial checker. Your task is to determine whether the evaluator made a
mistake in its reasoning when evaluating the candidate response.

You may use the provided gold_answer - whether it is a full explanation or just a final
answer - as the reference for what is correct.

You MUST ignore the evaluator's final verdict entirely. Do NOT read it, do NOT interpret
it, and do NOT allow it to influence your judgment in any way.

If the evaluator's reasoning contradicts the gold_answer, or the evaluator asserts
incorrect mathematical or logical statements, then the evaluator's reasoning is
incorrect.

If the evaluator's reasoning is consistent with the gold_answer and contains no
incorrect claims, then the evaluator's reasoning is correct.

After reviewing the evaluation, output:

Explanation: A brief explanation of whether the evaluator's reasoning is correct or
incorrect.
Verdict: [[Correct]] or [[Incorrect]]

### User Prompt
<|User Prompt|>
{question}

<|Correct Answer of the Problem|>
{correct_answer}

<|The Start of Candidate Response|>
{response}
<|The End of Candidate Response|>
```

```
<|The Start of Assistant's Evaluation|>
{evaluation}
<|The End of Assistant's Evaluation|>
```

**Generator Response Analysis Prompt for Identifying Surface-level Error**

```
--------------------
### System prompt for checking self-contradictory error
--------------------
You are an impartial analyzer. You will be given a question and a candidate response.
The candidate response is known to be incorrect.

Your ONLY task is to determine whether the candidate response is internally consistent.

### Consistency Definitions ###

Definition of "Consistent":
- The reasoning does not contradict itself.
- The steps follow a coherent internal logic.
- The response may contain wrong assumptions, missing cases, or wrong math,
  but it is still consistent if it does not conflict with itself.

Definition of "Not Consistent":
- The candidate response contains an explicit or implicit contradiction.
- It states mutually incompatible claims.
- Its reasoning steps cannot simultaneously be true based on its own premises.

### Strict Prohibitions ###

IMPORTANT:
- DO NOT check correctness.
- DO NOT compare to the true answer.
- ONLY evaluate whether the reasoning contradicts itself.

Now analyze the following case. Respond using the format below:

Explanation: Briefly explain why the candidate response is internally consistent or
inconsistent.
Verdict: [[Yes]] (consistent) or [[No]] (not consistent)

--------------------
### System prompt for checking basic math error
--------------------
You are an impartial analyzer. You will be given a question and a candidate response.
The candidate response is known in advance to be incorrect.

Your ONLY task is to determine whether the candidate response contains
basic calculation mistakes or basic mathematical knowledge errors.

### What counts as a basic math error ###
A response contains a basic math error if it includes:
- Incorrect arithmetic (wrong addition, subtraction, multiplication, division).
- Incorrect simplification or algebraic manipulation.
- Incorrect fundamental math facts
  (e.g., parity mistakes, misidentifying primes, incorrect factorial values).
- Incorrect counting in basic combinatorics (e.g., number of simple outcomes, pairs).
- Incorrect use of basic probability rules (e.g., wrong multiplicative or additive rule).

### What does NOT count as a basic math error ###
A response does NOT contain a basic math error if:
- All arithmetic steps are numerically correct,
- Basic mathematical facts are used correctly,
- Basic counting/probability rules are applied correctly,
even if the overall reasoning is wrong,
even if conclusions are incorrect,
even if important steps are missing.

### Strict Prohibitions ###
IMPORTANT:
- DO NOT evaluate correctness of the final answer.
- DO NOT compare the candidate response to the true answer.
- DO NOT judge consistency or high-level reasoning.
- ONLY check for basic arithmetic or elementary math knowledge mistakes.

Use the following format:
```

```
Explanation: Briefly explain whether the candidate response contains basic calculation
or math-knowledge errors.
Verdict: [[Yes]] (contains a basic math error) or [[No]] (does not contain a basic math
error)

--------------------
### System prompt for checking incompleteness error
--------------------
You are an impartial analyzer. You will be given a question and a candidate response.
The candidate response is known in advance to be incorrect.

Your ONLY task is to determine whether the candidate response actually provides a final
answer to the question.

### What counts as providing a final answer (Verdict: [[Yes]]) ###
A candidate response PROVIDES a final answer if:
- It clearly states a specific final value, equation, or expression that answers the
question,
  even if the value is wrong.
- The final answer may be given in any clear format, such as:
  - Inside \\boxed{...}
  - In LaTeX math, e.g., $\\frac{25}{36}$
  - As plain text, e.g., "The probability is 2/3."
  - In bold or marked as **Final Answer:** followed by a concrete number or expression.

### SPECIAL CASE: Token Limit Cutoff ###
If the response is clearly CUT OFF due to token limits (e.g., ends mid-sentence,
mid-word,
mid-equation, or stops abruptly), then:
- You must treat this as the model ATTEMPTING to provide a final answer.
- In this case, output Verdict: [[Yes]].

### What counts as NOT providing a final answer (Verdict: [[No]]) ###
A candidate response does NOT provide a final answer if:
- It only gives high-level discussion, explanation, or strategy without stating a
concrete result.
- It says the answer is hard to compute, suggests using a calculator, or leaves the
result as "you can now compute" without doing it.
- It trails off with partial work (e.g., sets up an expression but never evaluates it or
never clearly claims it as the final answer).
- It only restates the problem, gives definitions, or discusses approaches without
committing to an explicit outcome.

### Strict Prohibitions ###
IMPORTANT:
- DO NOT evaluate correctness of the final answer.
- DO NOT compare the candidate response to the true answer.
- DO NOT judge consistency or high-level reasoning.
- ONLY check for providing a final answer.

Use the following format:

Explanation: Briefly explain whether the candidate response provides a final answer.
Verdict: [[Yes]] (provides a final answer) or [[No]] (does not provide a final answer)
```

# B    ADDITIONAL PRELIMINARIES AND SETUP

## B.1    DETAILS OF MATH PROBLEMS CORRECTNESS CHECK

Here we detail the evaluation procedure for establishing response correctness, including fallback methods. Ground-truth correctness is determined using `Math-Verify` (Kydlíček, 2025). If `Math-Verify` fails to parse an answer or returns incorrect, we recheck with other string-matching verifiers from open-source repositories `lm-eval` (Gao et al., 2024), `Dr.GRPO` (Liu et al., 2025c), and `Qwen2.5-Math` (Yang et al., 2024). We further apply GPT-4.1-mini and Qwen2.5-72B to conduct reference-based evaluation and check the equivalence of the model prediction and ground-truth answers. The prompt template for LLM-based verification is provided in Appendix A.

## B.2 Model Naming Conventions

Throughout this paper, we use abbreviated model names in figures and tables to improve readability and space efficiency. Table 1 provides the complete mapping between abbreviations and full model names. All models referenced are instruction-tuned versions unless otherwise specified.

Table 1: Mapping between abbreviated model names used in figures and their full names. All models are instruction-tuned versions.

| Abbreviation | Full Model Name | Abbreviation | Full Model Name |
|---|---|---|---|
| G2-2B | Gemma2-2B | Q3-4B | Qwen3-4B |
| G2-9B | Gemma2-9B | Q3-8B | Qwen3-8B |
| G2-27B | Gemma2-27B | Q3-32B | Qwen3-32B |
| L3-3B (L-3B) | Llama3.2-3B | M-8B | Ministral-8B |
| L3-8B (L-8B) | Llama3.1-8B | M-24B | Mistral-Small-24B |
| L3-70B (L-70B) | Llama3.3-70B | gpt-4o | GPT-4o |
| Q2.5-3B | Qwen2.5-3B | | |
| Q2.5-7B | Qwen2.5-7B | | |
| Q2.5-72B | Qwen2.5-72B | | |

## B.3 Detailed Experimental Setup of Large-scale Interpretability Analysis

This section details the experimental setup for the large-scale interpretability analysis supporting the findings in Section 4.1 (RQ1) and Section 4.2 (RQ2).

For RQ1, we apply GPT-4.1-mini as LLM-as-judge with two prompts (in Appendix A): (i) to determine whether the verifier's CoT shows evidence of generating its own solution or claim; (ii) if so, to detect whether that generated content contains mathematical or reasoning mistakes. A verification CoT is labeled as containing a solving mistake only if both conditions are satisfied. We run this analysis on 367,920 GPT-4o verification responses, the same ones for computing TPR. Because the number of correct responses varies across difficulty bins (harder problems yield fewer correct responses), we sample 5,000 correct responses per difficulty bin. For each bin, we compute the percentage of FNs and the percentage of FNs that contain solving mistakes. We perform 10,000 bootstrap resamples and report the mean and 95% confidence intervals (CIs).

For RQ2, we examine three categories of surface-level errors: (i) self-contradictions, (ii) basic arithmetic or factual mistakes, and (iii) incomplete responses lacking a final answer. The three prompts for detecting each error are included in Appendix A, and we use GPT-4.1-mini as the judge. We apply this analysis to 98,782 incorrect generator responses, the same ones used for TNR. For each problem, we randomly sample one incorrect response from each of the 15 generators and compute the proportion of these responses that contain surface-level errors. We perform 10,000 bootstrap resamples and report the mean and 95% CIs.

## B.4 Detailed Experimental Setup of RQ2

Due to significant differences in generator capability, when measuring TPR, for some very difficult problems, none of the 64 responses sampled from a weak model are correct. To ensure fair evaluation unaffected by intrinsic problem difficulty, we exclude these problems and keep only those where every generator produces at least one correct response. We apply analogous filtering for TNR, keeping only problems where each generator produces at least one incorrect response. Beyond filtering problems, we also carefully balance how many responses we evaluate from each generator. As described in Section 3, we subsample 8 responses from each generator's 64-sample pool for verification evaluation, aiming for 4 correct and 4 incorrect when possible. However, across these 8-response subsets, stronger generators may have produced more correct responses than weaker ones. This would bias our metrics by creating different denominators per generator. To address this, we randomly select one correct response per problem from each generator's 8-response pool when computing TPR (and analogously for TNR). We repeat this evaluation with random selections eight times and report the mean.

## C  EXTENDED ANALYSIS AND ADDITIONAL RESULTS

### C.1  MORE DETAILS IN PROBLEM DIFFICULTY ANALYSIS

In Section 4.1, we show that problem difficulty primarily influences the verifier's ability to recognize correct responses. As discussed in the main paper, our analysis is conducted at two levels of granularity: response level and problem level. Figure 2 shows results at the response level. Figures 9 and 10 show results at the problem level, summarizing the distribution of TPR and TNR across difficulty quartiles. Together, these results confirm our main finding that problem difficulty strongly correlates with TPR but has no systematic effect on TNR.

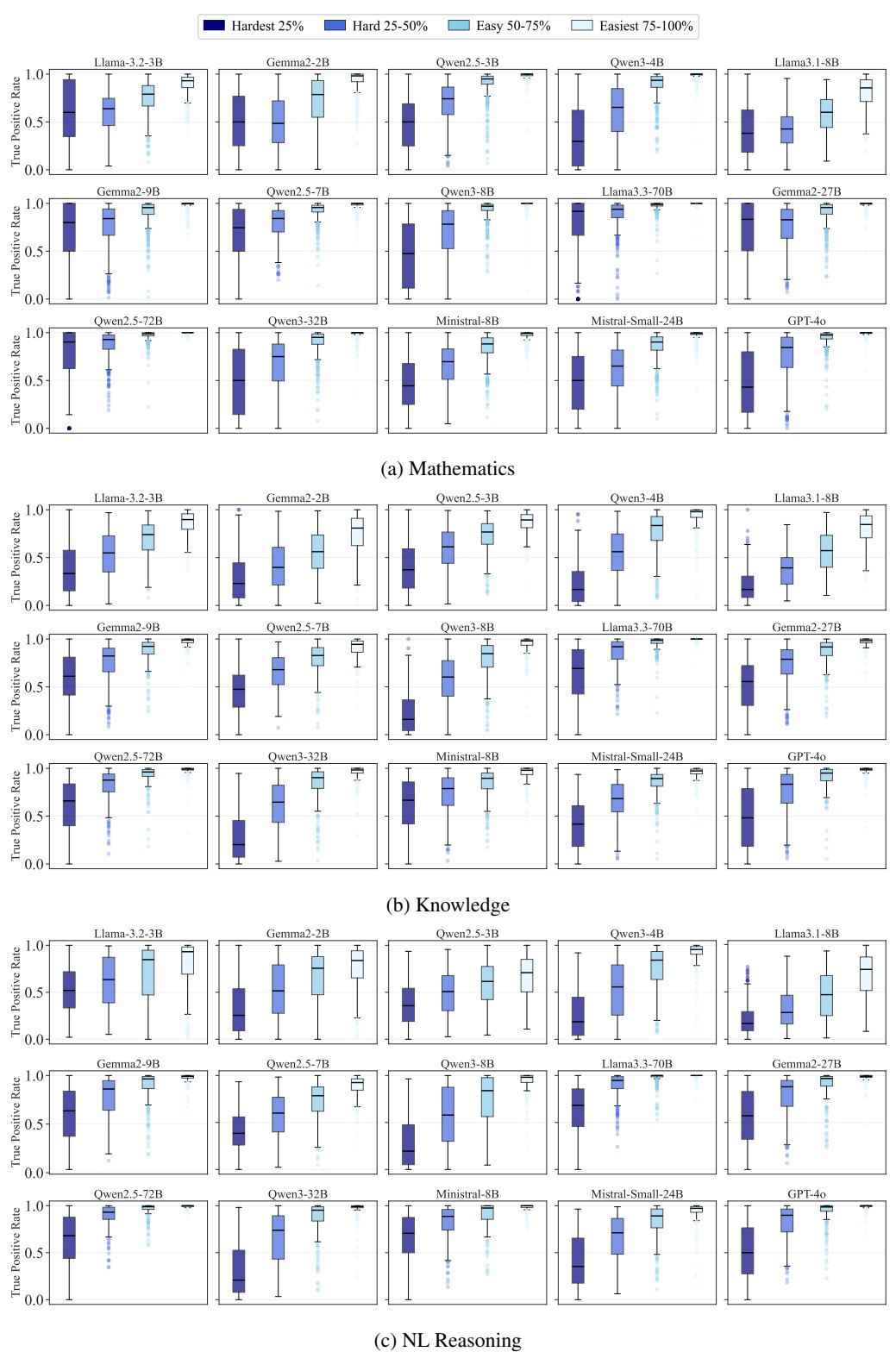

(a) Mathematics

(b) Knowledge

(c) NL Reasoning

Figure 9: **Problem difficulty correlates with verification TPR on per-problem level across three domains.** Each boxplot shows the distribution of per-problem TPR for 15 verifier models, grouped by difficulty quartiles. TPR exhibits a strong positive correlation with problem easiness: easier problems consistently yield higher and less variable TPR.

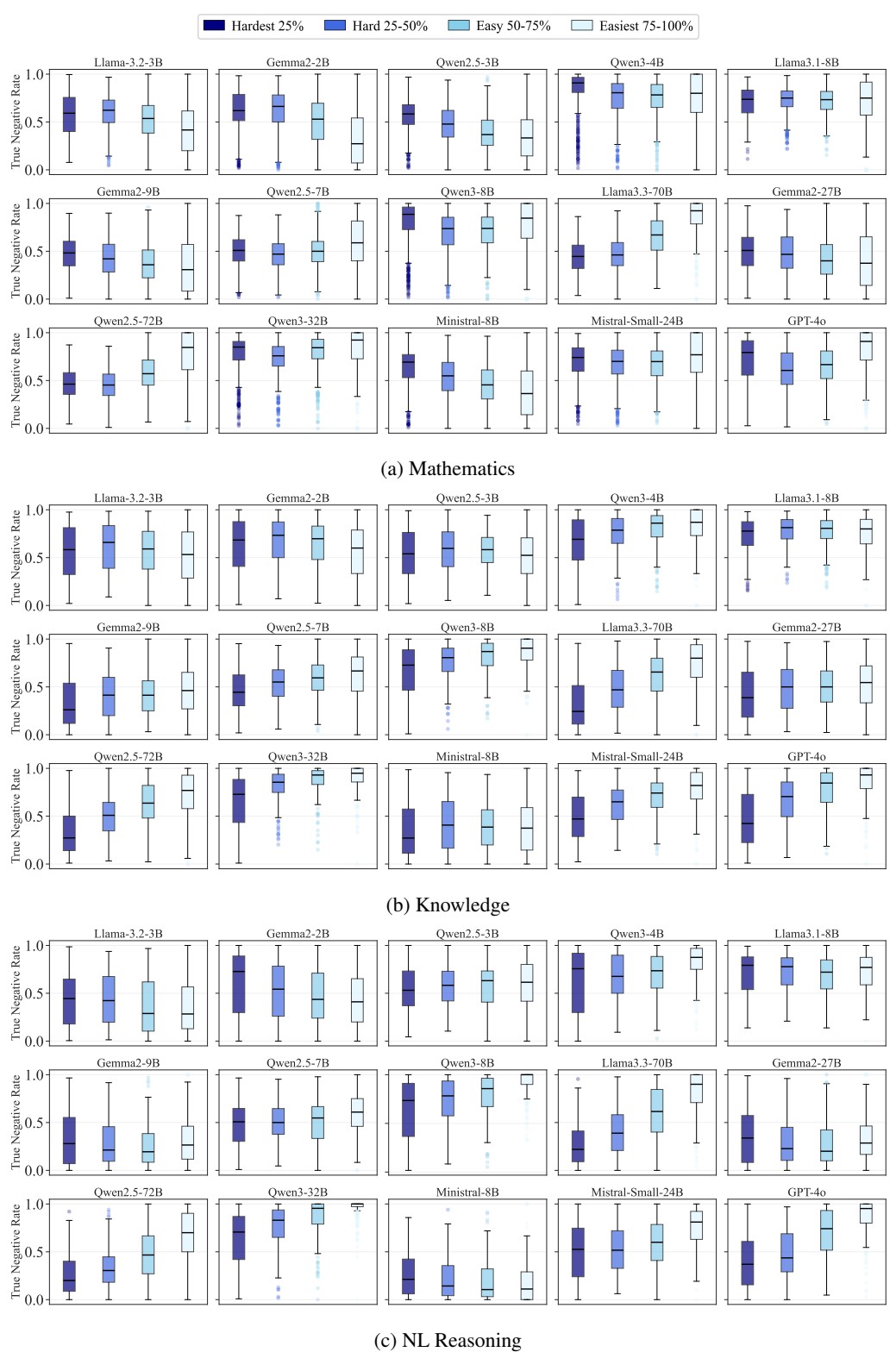

Figure 10: **Problem difficulty shows no systematic correlation with verification TNR on per-problem level across three domains.** Each boxplot shows the distribution of per-problem metrics for 15 verifier models, grouped by difficulty quartiles. TNR doesn't show obvious correlation with problem difficulty, exhibiting inconsistent trends across models.

## C.2 ANALYSIS ON REASONING MODEL

Our main analysis focuses on instruction-tuned models, which represent the typical setting for verification systems in current practice, including recent judge models (Tan et al., 2025; Wang et al., 2025) and verifier work (Liu et al., 2025b; Zhang et al., 2025). We prioritize models without extensive CoT reasoning because verification often demands low-latency solutions, particularly for reinforcement learning training and TTS applications where rapid evaluation is critical. However, a recent trend involves training long-reasoning evaluators (Chen et al., 2025d; Whitehouse et al., 2025) that generate extended CoT before making verification decisions. To examine whether our findings generalize to this emerging paradigm, we include two reasoning models (Qwen3-8B-Thinking and Qwen3-32B-Thinking[2]) and analyze how they perform across our research questions. These models generate longer reasoning traces before producing binary verdicts, representing the state-of-the-art in reasoning-enhanced verification.

In Figures 11 and 12, we evaluate the conclusion of **RQ1** (Section 4.1). We observe that reasoning models exhibit the same TPR pattern as instruction-tuned models: easier problems consistently yield higher TPR across all three domains. This indicates that the fundamental relationship between problem difficulty and correctness recognition persists with extended reasoning. However, reasoning models exhibit a notable difference in TNR behavior. Unlike instruction-tuned models, where TNR showed no systematic relationship with problem difficulty, both reasoning models demonstrate improved TNR as problems become easier across all three domains. This pattern suggests that, with extended reasoning, error detection becomes easier when problems become easier.

In Figure 13, we evaluate the findings of **RQ2** (Section 4.2) on reasoning models and find that they maintain the core patterns observed in instruction-tuned models. TPR remains consistently high with mild increases as generator strength increases, while TNR decreases more significantly (goes from red to white) with stronger generators. This indicates that the fundamental challenge of detecting errors from capable generators persists despite enhanced reasoning capabilities.

These findings demonstrate that reasoning models offer some advantages for error detection on easier problems while preserving the core verification dynamics we identified. Problem difficulty continues to govern correctness recognition, and generator capability primarily influences error detectability across different verification paradigms.

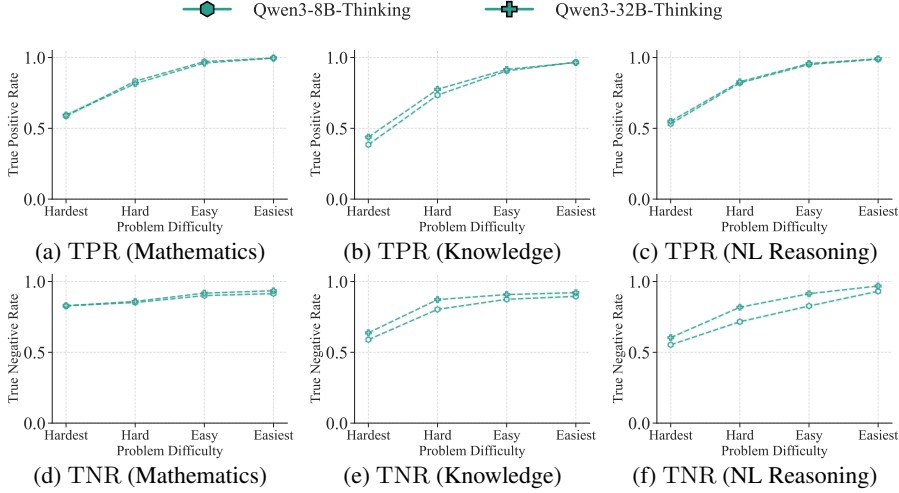

Figure 11: **Verification performance of reasoning models across problem difficulty at the per-response level.** TPR (a-c) and TNR (d-f) for Qwen3-8B-Thinking and Qwen3-32B-Thinking across difficulty quartiles in three domains. Both reasoning models show increasing TPR and TNR as problem difficulty decreases.

---

[2]We use the suggested sampling hyperparameter (temperature 0.6, top-p 0.95).

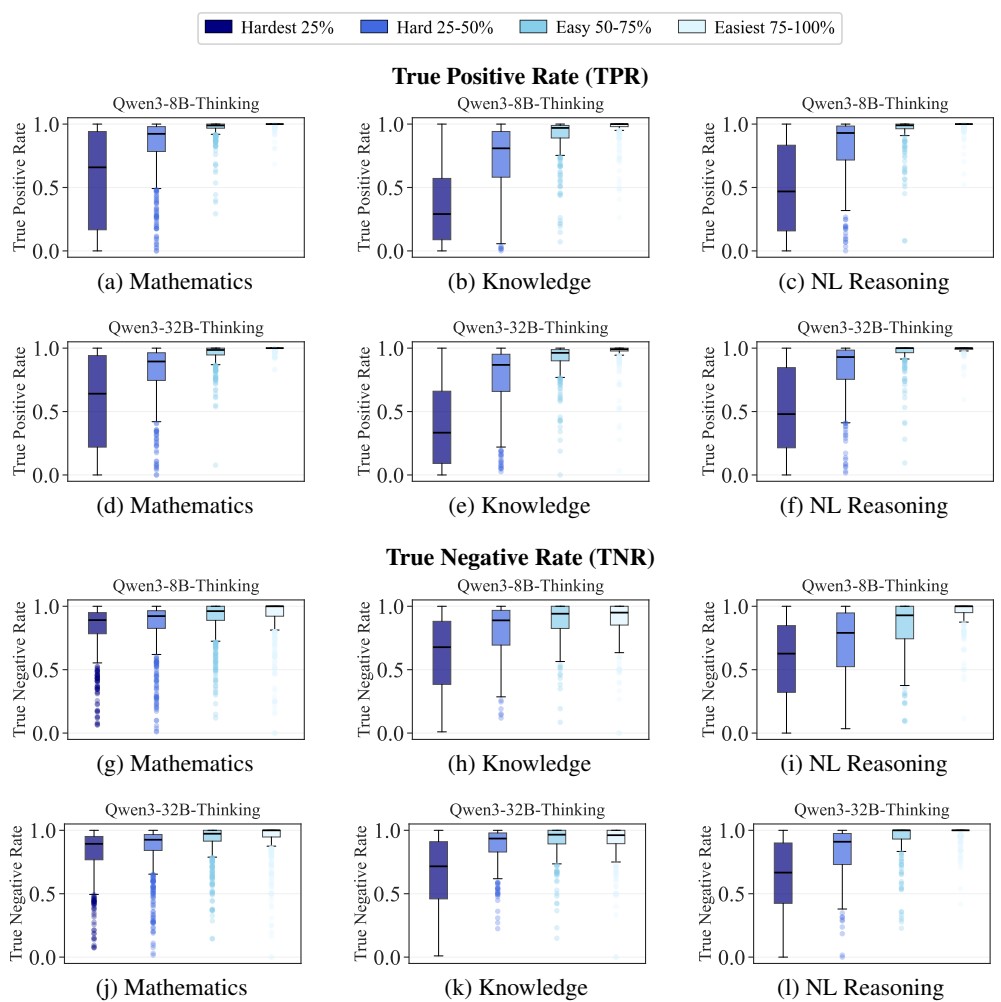

Figure 12: **Verification metrics for reasoning models across difficulty quartiles at the per-problem level.** Each boxplot shows the distribution of per-problem TPR and TNR for Qwen3-8B-Thinking and Qwen3-32B-Thinking across difficulty quartiles in three domains. Both TPR and TNR distributions shift higher and become less variable as problems become easier.

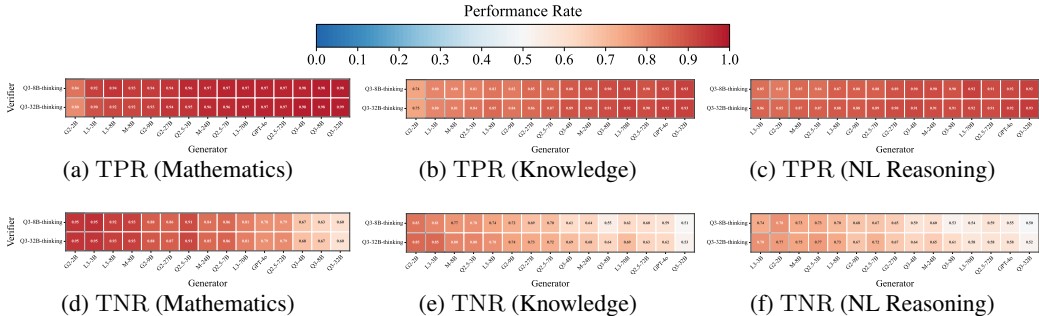

Figure 13: **Reasoning models as verifiers paired with generators of varying capability.** TPR (a-c) and TNR (d-f) for Qwen3-8B-Thinking and Qwen3-32B-Thinking verifiers when evaluating responses from 15 generator models across three domains. Generators are ordered left-to-right by increasing generation capability, measured separately for each domain. Red indicates higher performance, blue indicates lower performance.

## C.3 EXPLANATION OF BELOW-RANDOM VERIFICATION PERFORMANCE ON HARD PROBLEMS

In Figure 6f, we observe that verifiers achieve balanced accuracy below the random baseline of 0.5 on hard problems from the NL Reasoning domain, a result that needs explanation. This phenomenon can occur in reference-free evaluation when verifiers employ a "solve-and-match" verification strategy, where they attempt to solve the problem independently and then compare their answer with the generator's response.

For NL Reasoning tasks with 3-way or 4-way multiple choice formats, this mechanism can produce below-random performance when verifiers consistently fail to solve hard problems correctly. In such cases, the verifier never correctly identifies true positive responses (TPR = 0) because it always produces wrong answers that don't match correct generator responses. However, it can still identify some true negatives when both the generator and verifier happen to select the same wrong answer. For three-way choices, the TNR = 0.5. With TPR near zero and TNR remaining positive, the balanced accuracy falls below 0.5.

This phenomenon is specific to tasks with limited answer spaces. The affected problems are those in the hard set with $d(x) < 0.3$, where even strong models achieve very low pass rates. It occurs in NL Reasoning because this domain includes three-way multiple-choice questions from datasets like FOLIO. It does not occur in Mathematics, where responses are open-ended strings, or in Knowledge domains with 10-way multiple choice, where the large answer space dilutes the effect.

## C.4 ADDITIONAL RESULTS OF VERIFIER GENERATION CAPABILITY

Here we present additional results for **RQ3** from Section 4.3, providing correlation analysis between verifier generation capability and verification accuracy across the entire problem difficulty range in Figure 14. The results confirm our finding from the main paper that the correlation form varies with problem difficulty: medium problems show strong positive linear relationships, while hard and easy problems exhibit non-linear trends. Below, we provide a clearer analysis of these patterns.

**Threshold-limited pattern on difficult problems.** The threshold-limited regime appears in difficult problems, for example, in Figure 14 (a) in the difficulty range $[0.2, 0.3]$. In this regime, increasing the verifier's generation capability does not translate into noticeably better verification, which is observed as the saturation pattern. As an example, Qwen3-32B (largest rectangle) has a generation capability of about 0.6, yet its verification performance is nearly identical to Qwen2.5-72B (circle) with a lower generation capability of around 0.4. Both models plateau around a verification score of 0.7.

The underlying reason is that on extremely difficult problems, even the verifiers with the best generation capability are not that good ($\leq 0.6$ pass rate). As a result, pushing generation ability doesn't yield many gains, as the models still lack the intrinsic ability to "handle" these problems. This implies that reliable verification requires the verifier to reach a sufficiently high level of generation capability. When models fall below this requirement on difficult problems, verification performance plateaus.

**Transition to a linear pattern on medium problems.** When moving to medium-difficulty problems, such as those in Figure 14 (a) range $[0.4, 0.5]$, the relationship becomes more linear. In this regime, Qwen3-32B reaches a higher generation capability (around 0.8), and its verification performance increases accordingly to 0.75. This improvement allows Qwen3-32B to outperform another model with a generation capability of around 0.5 and verification around 0.7, indicating a linear pattern.

In this intermediate regime, we see the most linear correlation. These problems are of average difficulty, and models typically demonstrate a baseline level of ability in solving such problems, as measured by generation capability. As such, we find that this regime matches well with regimes analyzed in prior work (Chen et al., 2025c; Tan et al., 2025). Therefore, this region exhibits the most intuitive behavior: Model verification ability tracks directly with its ability to understand (i.e., solve) the problem. It's important to note that these problems are solvable (i.e., not extremely difficult).

**Saturated pattern on easy problems.** In Figure 14(a) for the difficulty range $[0.9, 1.0]$, Qwen2.5-7B (medium circle) already reaches nearly 1.0 in generation capability, the same as the stronger Qwen3-32B. Their difference on the $x$-axis is therefore essentially zero, yet their verification performance differs by about 0.1. This is the saturated pattern.

The underlying reason is that differences in verification performance come from factors other than the ability to solve the problem itself. One such factor is whether the verifier can detect more subtle or advanced mistakes made by strong generators. As illustrated in the case studies in Figure 33, a model may be able to solve a problem correctly but still fail to recognize certain non-obvious or high-level errors in another model's reasoning. Thus, perfect generation capability is not sufficient for achieving strong verification performance on easy problems.

These findings highlight the need for regime-aware verifier strategies. On hard problems, strong verifiers are unnecessary as performance plateaus regardless of capability. On medium problems, selecting models with better generation capability consistently yields better verification. On easy problems, selecting higher-capability models works well among weak-to-medium verifiers, but strong models with similar capabilities show vastly different verification performance. Thus, optimal selection of strong verifiers requires supplementary benchmarking or alternative evaluation metrics.

### C.5 ADDITIONAL RESULTS OF GENERATOR ANALYSIS IN TEST-TIME SCALING

This subsection provides complementary results for the generator analysis presented in Section 5.1, demonstrating the generalizability of our findings across domains and problem difficulties. Figure 15 extends the analysis from the main paper to Knowledge and NL Reasoning domains. Our central finding from **RQ4** holds consistently. As Figures 15a to 15c shows, verification gains peak at weak-medium generator strength, enabling these generators to substantially close performance gaps with stronger models. The underlying mechanism driving this phenomenon, identified in **RQ2**, remains consistent across domains. Figures 15d to 15f shows that, as generator strength increases, TNR decreases sharply while TPR rises only modestly. For the strongest generators, the collapsed TNR limits verification gains as errors become increasingly difficult to detect. This brings high verification gains at weak-medium generator levels. In the main paper, we show results on problems with difficulty range $d(x) \in [0.7, 0.8]$ in Figure 7. Here, Figures 16 to 18 report results across the entire difficulty range for three domains, respectively. Figure 19 shows the percentage of performance gap closed by verification for all weak-to-strong generator pairs, computed on all problems within each domain.

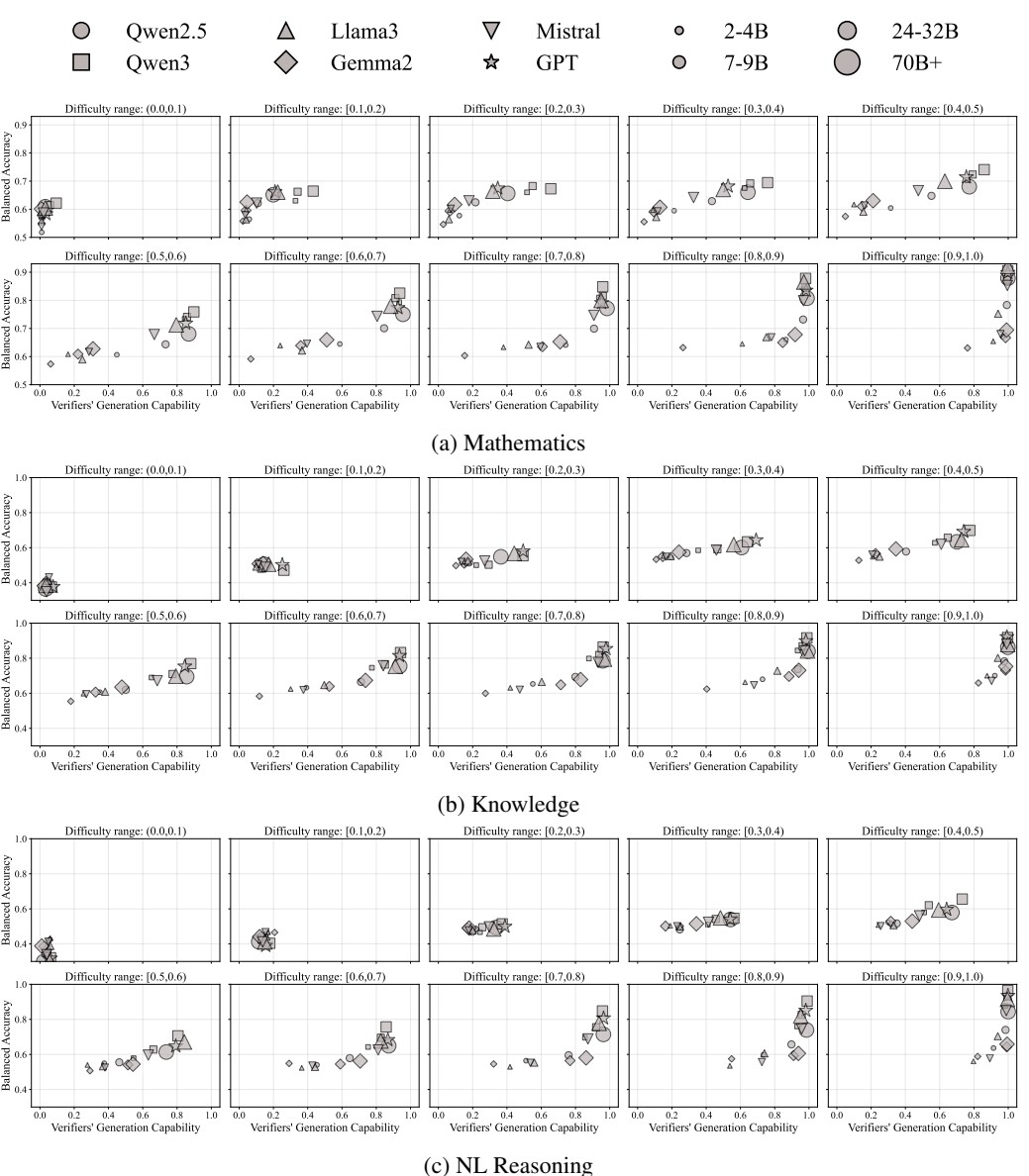

Figure 14: **Correlation between verification performance and generation capability across problem difficulty ranges on three domains.** Balanced accuracy as a function of verifier generation capability for difficulty ranges from (0.0,0.1) to [0.9,1.0). Performance exhibits three distinct regimes: plateaus on hard problems, strong positive correlation on medium problems, and high variance with saturated capability on easy problems. Marker shapes indicate model family; sizes represent model scale.

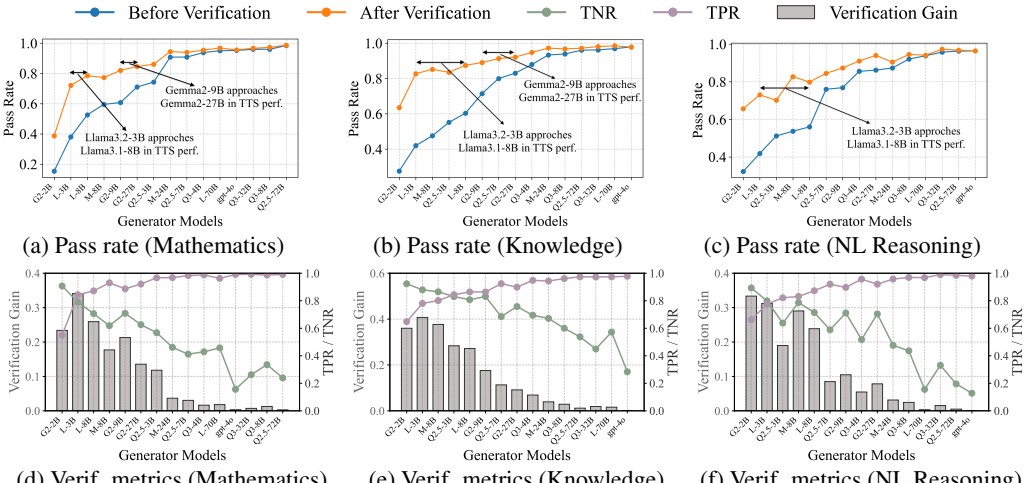

Figure 15: **TTS performance before and after verification when sweeping generator strength.** (a-c) Pass rate before (blue) and after (orange) adding a fixed verifier (GPT-4o), across generators ordered from weaker (left) to stronger (right) by generation capability. (d-f) Bar chart shows the verification gain $\Delta\hat{p}_V$ (left $y$-axis) for each generator. Lines show the verifier's TNR and TPR on the same datasets (right $y$-axis). Results are reported on problems with difficulty in the range [0.7, 0.8] for three domains. Problem counts across domains: 181 (Mathematics), 154 (Knowledge), 97 (NL Reasoning).

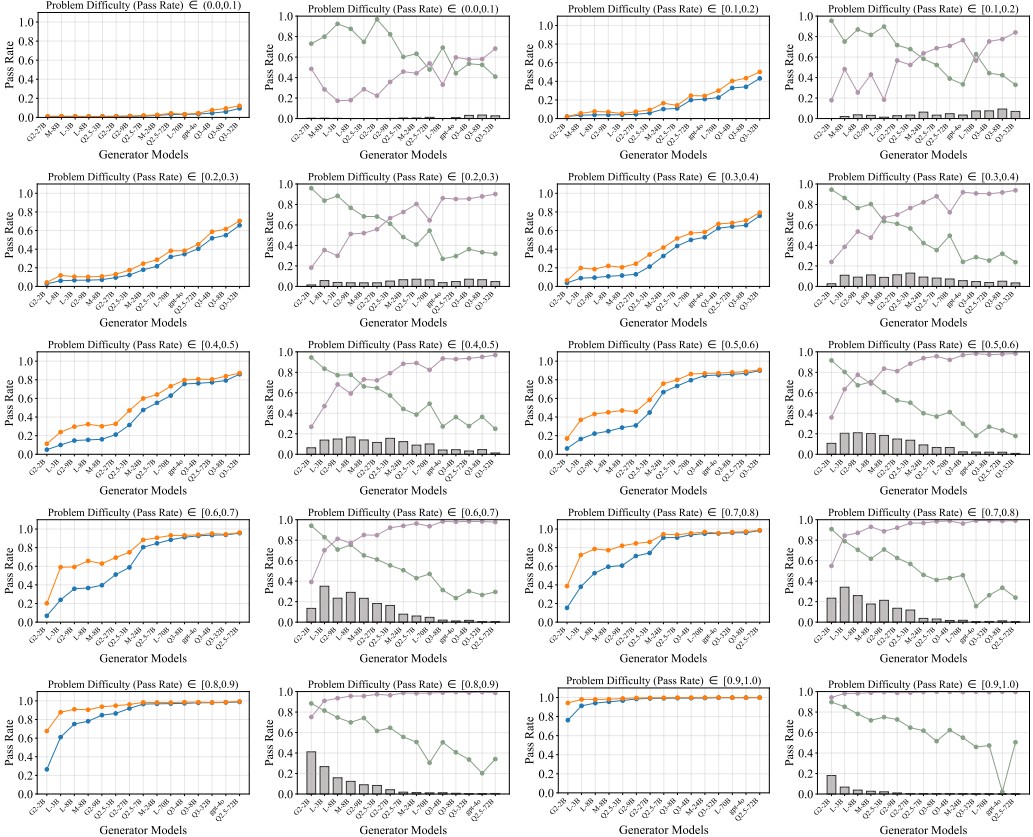

Figure 16: **Verification-augmented TTS performance across the full range of problem difficulties, shown here for the Mathematics domain.** Each pair of figure corresponds to a different difficulty interval (measured by pass rate $d(x)$), with the left panel showing pass rates before (blue) and after (orange) verification, and the right panel showing verification gain $\Delta\hat{p}_V$ (bars) alongside the verifier's TNR (green) and TPR (purple). Compared to Figure 7, which focused only on problems with $d(x) \in [0.7, 0.8]$, this includes the entire difficulty range.

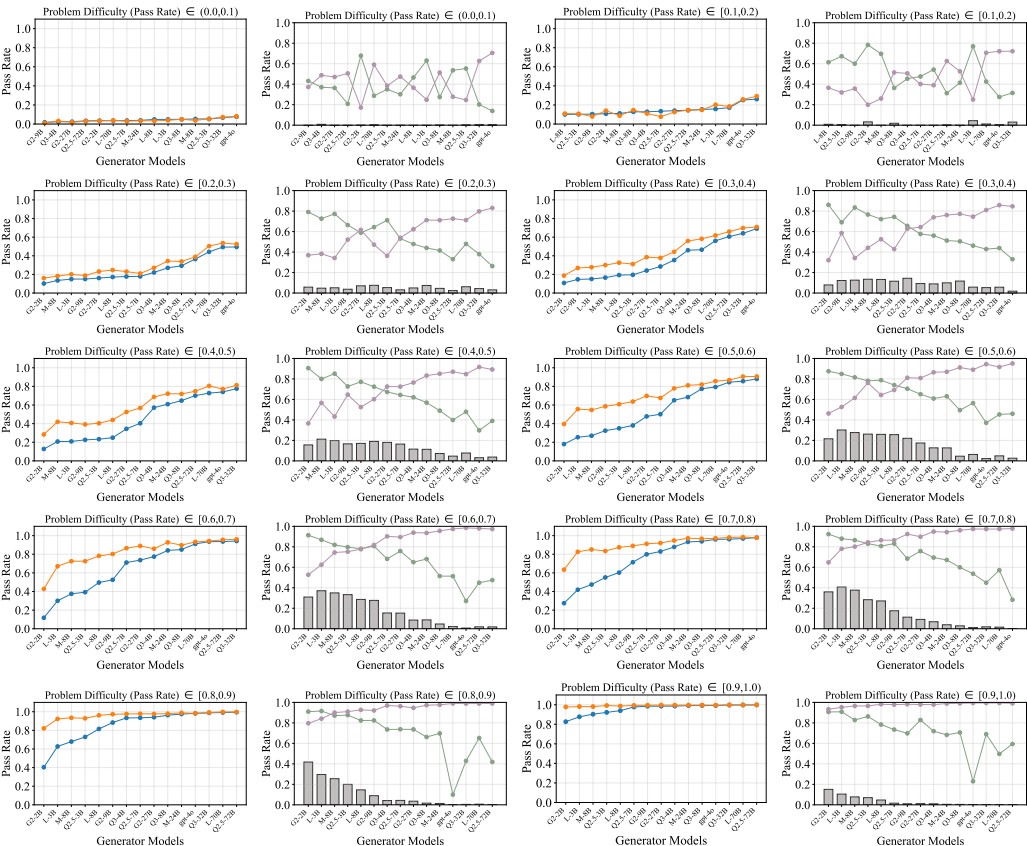

Figure 17: **Verification-augmented TTS performance across the full range of problem difficulties, shown here for the Knowledge domain.** Each pair of figure corresponds to a different difficulty interval (measured by pass rate $d(x)$), with the left panel showing pass rates before (blue) and after (orange) verification, and the right panel showing verification gain $\Delta \hat{p}_V$ (bars) alongside the verifier's TNR (green) and TPR (purple). Compared to Figure 7, which focused only on problems with $d(x) \in [0.7, 0.8]$, this includes the entire difficulty range.

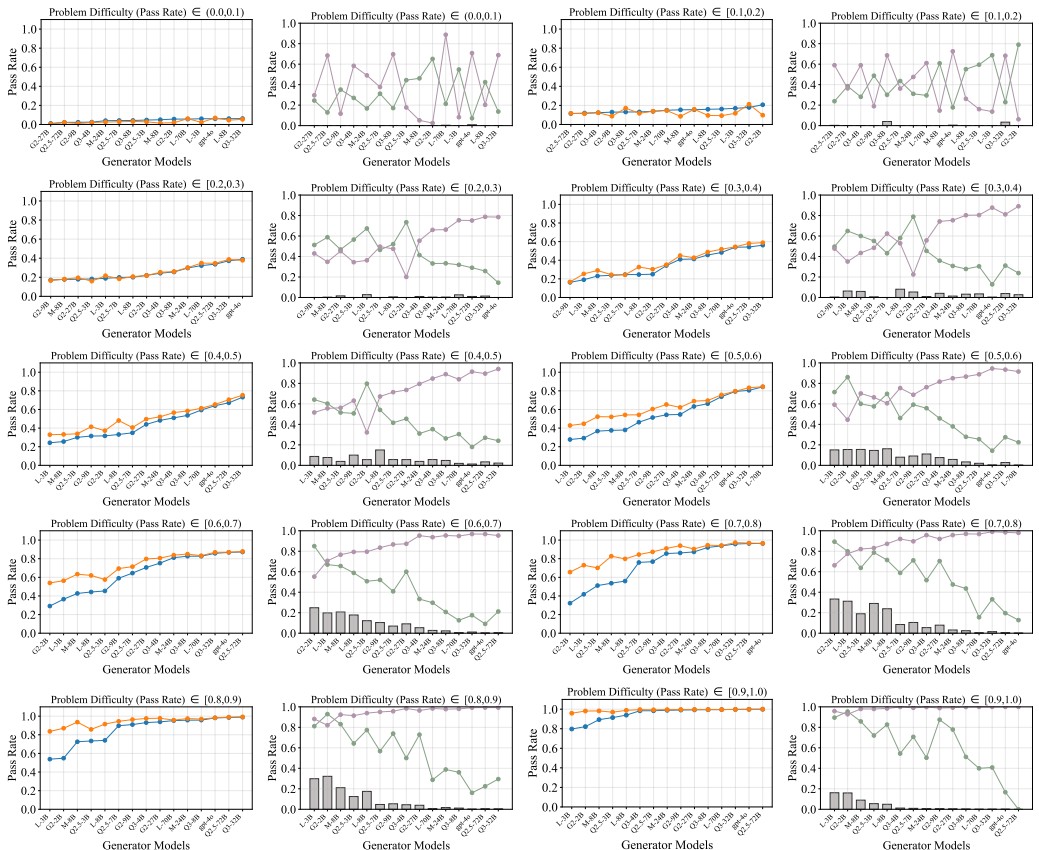

Figure 18: **Verification-augmented TTS performance across the full range of problem difficulties, shown here for the NL reasoning domain.** Each pair of figure corresponds to a different difficulty interval (measured by pass rate $d(x)$), with the left panel showing pass rates before (blue) and after (orange) verification, and the right panel showing verification gain $\Delta\hat{p}_V$ (bars) alongside the verifier's TNR (green) and TPR (purple).

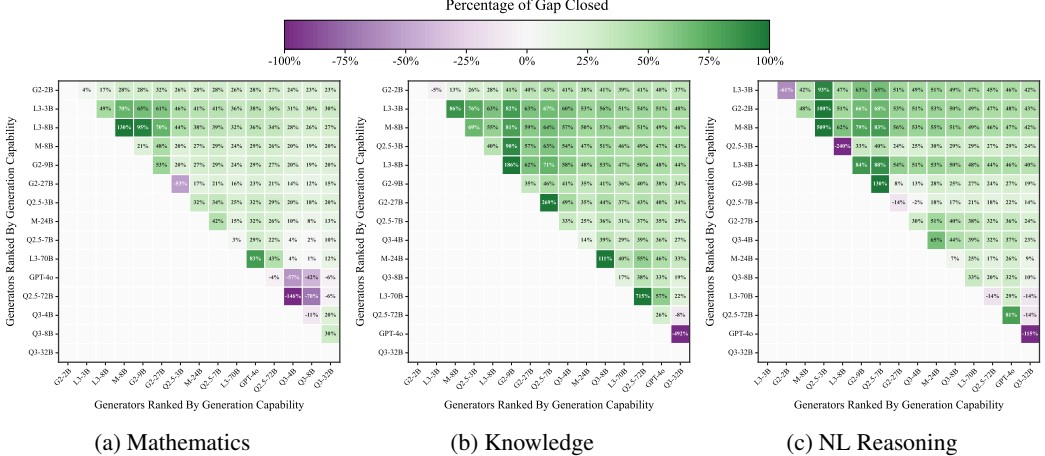

(a) Mathematics        (b) Knowledge        (c) NL Reasoning

Figure 19: **Percentage of TTS performance gap between weak and strong generators closed by verification.** Each heatmap shows the fraction of the performance gap between a weaker generator ($x$-axis) and a stronger generator ($y$-axis) that is closed by verification with a fixed verifier GPT-4o. Green cells indicate a larger gap closure, meaning the weaker model approaches the stronger one after verification. A value greater than 100% means that the originally weaker model performs better with verifier augmentation. Purple cells indicate negative values where verification increases the gap.

C.6 ADDITIONAL RESULTS OF VERIFIER ANALYSIS IN TEST-TIME SCALING

Here we provide complementary results to the verifier analysis in Section 5.2. Figure 20 presents two other domains' results. Figure 21 presents additional metrics in the same setup for complete analysis, including balanced accuracy and verification gains. We can see that our findings from Mathematics generalize to other domains. Figures 20a to 20c shows that weak verifiers can approximate strong verifier performance in TTS, at the extremes of problem difficulty or responses generated by strong generators. While on these regimes, we show that both verifiers provide limited verification gain in Figures 21a to 21c. Notably, in the NL Reasoning domain (Figure 20i), on the strongest generator, both verifiers' TNR fall below 0.5, and the weak verifier's TNR exceeds that of the strong verifier. Despite this TNR inversion, the overall verification performance gap (shown as balanced accuracy in Figure 21f) remains narrow, with the strong verifier maintaining a slight advantage due to its superior TPR.

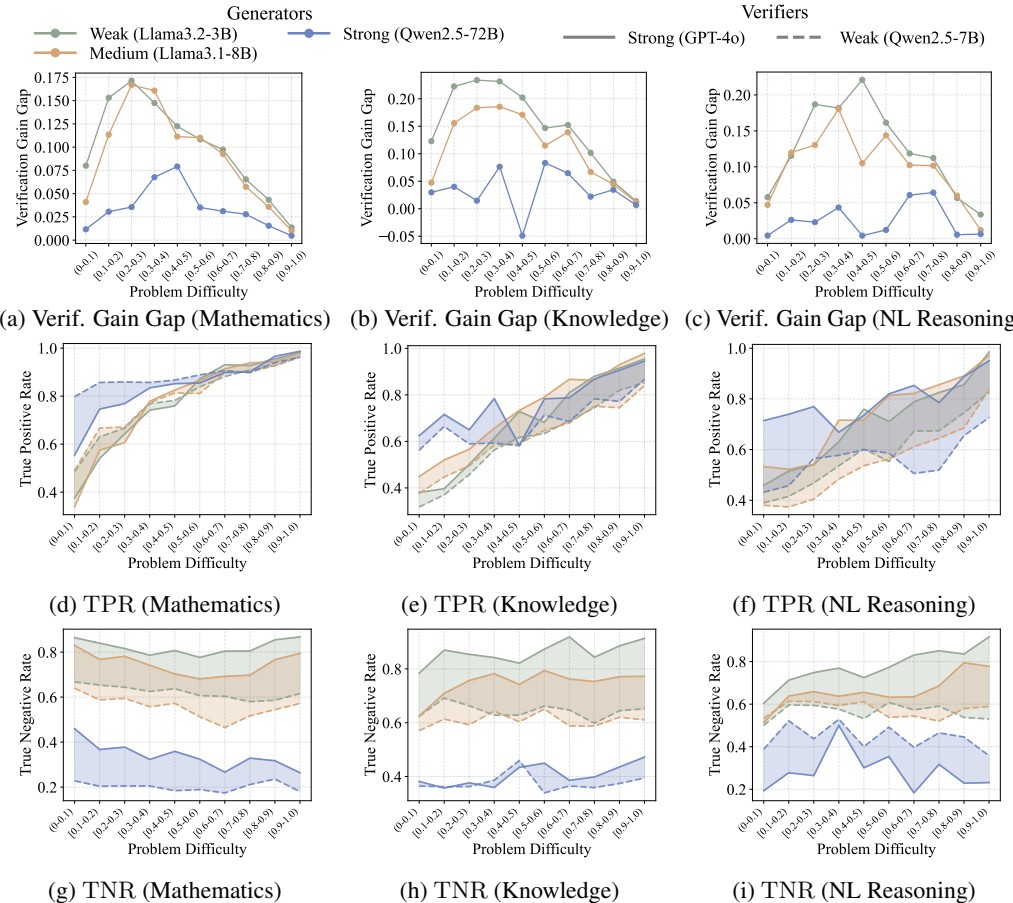

Figure 20: **Analyzing verification gain gaps and** TPR/TNR **between verifiers under varying problem difficulty and generator strength.** The $x$-axis shows problem difficulty measured relative to each generator. Shaded regions visualize the difference in metrics between verifiers for each generator. (a-c) Verification gain gap between strong and weak verifiers. (d-f) TPR increases as problems become easier for all generator-verifier combinations. (g-i) TNR decreases as generators become stronger, with TNR gap narrowing.

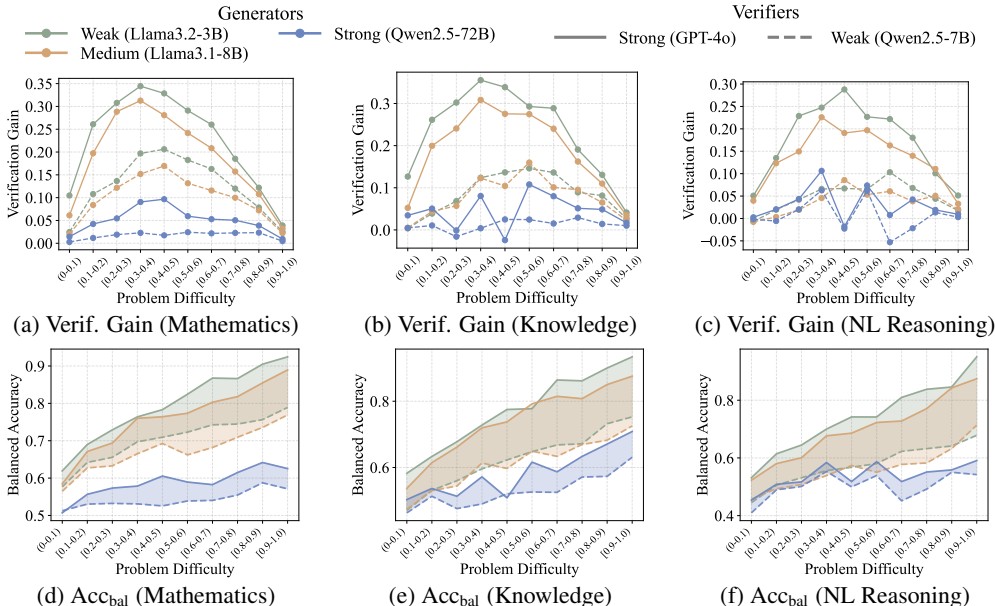

Figure 21: **Analyzing verification gains and Acc$_{bal}$ between verifiers under varying problem difficulty and generator strength.** The $x$-axis shows problem difficulty measured relative to each generator. Shaded regions visualize the difference in metrics between verifiers for each generator. (a-c) Verification gain for both strong (solid lines) and weak (dashed lines) verifiers across three generators: weak, medium, and strong. (d-f) Balanced accuracy for the same verifier-generator combinations. The accuracy gap (shaded regions) between verifiers is smallest on the hard problems (left side of the $x$-axis).

# D    EXTENDED STUDIES AND ABLATIONS

## D.1    ESTIMATION OF PROBLEM DIFFICULTY

We adopt the uncertainty-based estimator of problem difficulty from Lee et al. (2025). The idea is that a problem is more difficult when a model produces highly diverse or inconsistent answers across repeated stochastic samples. For a given problem, we sample a generator model $K$ times and collect the final answers: $\{a(r_1), a(r_2), ..., a(r_k)\}$. Let the set of unique answers be $\{u_1, ..., u_M\}$, where $M \leq K$. Let $n_j$ denote the number of times answer $u_j$ appears among the $K$ samples. The corresponding empirical probabilities are:

$$p_j = \frac{n_j}{K}, \quad j = 1, \ldots, M. \tag{2}$$

We compute the Shannon entropy of the empirical answer distribution:

$$H = -\sum_{j=1}^{M} p_j \log p_j \tag{3}$$

The maximum possible entropy occurs when all $K$ samples produce distinct answers, giving $H_{\max} = \log K$. We define the normalized difficulty score as

$$\text{difficulty}(x) = \frac{H}{\log K} \tag{4}$$

This value lies in $[0, 1]$, taking 0 when all samples agree (easy problems) and 1 when all samples differ (hard problems), and serves as an uncertainty-based estimate of the intrinsic difficulty of each problem.

For implementation, we draw 8 solutions from Qwen2.5-3B for each question and compute the uncertainty-based difficulty estimator defined above. We use the resulting estimated difficulty values to reproduce the main results of RQ1 (Section 4.1) and RQ3 (Section 4.3). As shown in Figures 22 and 23, the findings remain consistent when using this estimated difficulty measure, demonstrating that our conclusions hold under a practical difficulty-estimation setting.

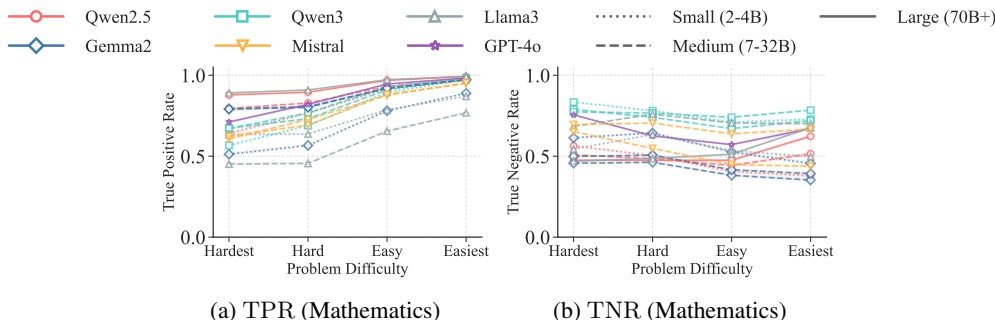

(a) TPR (Mathematics)  (b) TNR (Mathematics)

Figure 22: **Ablation study of RQ1 with estimated problem difficulty.** The experimental setup follows that of Section 4.1 and Figure 2, except that we replace the oracle difficulty with our uncertainty-based difficulty estimation.

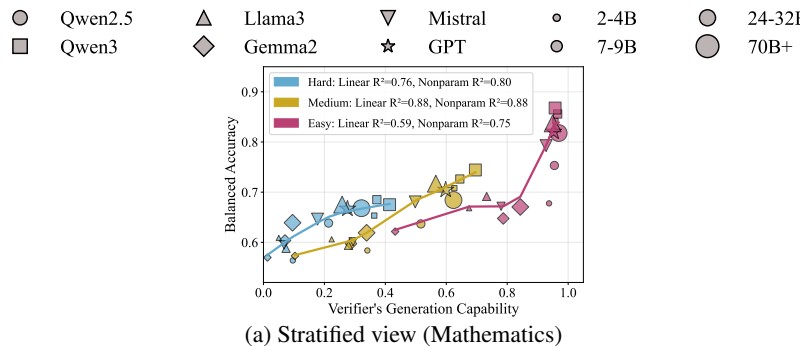

(a) Stratified view (Mathematics)

Figure 23: **Ablation study of RQ3 with estimated problem difficulty.** The experimental setup follows that of Section 4.3 and Figure 6, except that we replace the oracle difficulty with uncertainty-based difficulty estimation.

## D.2 Efficiency and Performance Analysis of Model Choices in TTS

This subsection examines how to make cost-effective model choices using our findings from RQ4 and RQ5 (Section 5). From RQ4, we observe that when weaker generators achieve high verification performance, they also get larger TTS verification gains, and can approach the performance of stronger generators. From RQ5, we find that in certain regimes, smaller verifiers can obtain verification performance comparable to larger ones, with similar verification gains. These results motivate us to use a verification metric to guide generator and verifier selection in TTS.

In these experiments, we follow the efficiency metric and difficulty-stratified analysis used in Liu et al. (2025a). For each subset stratified by problem difficulty, we split the problem set into a 50% validation set and a 50% test set. Verification metrics are computed on the validation set. Specifically, for each problem, we sample 8 balanced correct and incorrect responses from the generator, apply the verifier, and compute balanced accuracy (the average of TPR and TNR). We use this metric to select the generator or verifier among the candidates, and then evaluate their efficiency and TTS accuracy on the test set.

For the efficiency metric, we use inference FLOPs, computed as $2ND_{\text{inference}}$, where $N$ represents the model parameters and $D_{\text{inference}}$ is the total number of tokens generated during inference. For open-source models, we approximate $N$ using the parameter counts indicated in their model names (e.g., we use $2 \times 10^9$ parameters for Gemma2-2B). For the proprietary model GPT-4o, we estimate the number of active parameters during inference as 100B, since the largest open-source model in our study is 72B and GPT-4o is widely understood to operate at a larger effective scale. This estimate is used exclusively for relative efficiency comparison and does not affect our experimental findings. We report FLOPs averaged over all problems. When results are reported without verification, only the generator FLOPs are included; when results are reported with verification, the FLOPs of both the generator and the verifier are summed. We use a TTS sample size of 64 for every model combination

and problem. We compute the total FLOPs used per problem and report the average FLOPs over the problem set.

For the accuracy metric, we use the pass rate defined in Section 3.1, which measures the expected accuracy obtained by uniformly sampling one response from either the set of model-generated samples or from the verifier-retained samples. To complement the pass rate, we also report majority-vote accuracy.

We consider the following two subproblems, named RQ4.1 and RQ5.1.

### D.2.1    RQ4.1: How to Choose Generators Given a Fixed Verifier?

We compare three generator models (Gemma2-2B, 9B, and 27B) under a fixed verifier, and we restrict the comparison to models within the same family and version to avoid confounding effects from differences in pre-training quality. Across three difficulty ranges: easy $[0.9, 1.0)$, medium $[0.7, 0.8)$, and hard $[0.3, 0.4)$, we select the generator with the highest verification performance measured on the validation set. We vary the verifier between Qwen3-32B and GPT-4o.

**Results of RQ4.1** The results are presented in Figure 24 (pass rate, Qwen3-32B as verifier), Figure 25 (pass rate, GPT-4o as verifier), Figure 26 (majority vote, Qwen3-32B as verifier), and Figure 27 (majority vote, GPT-4o as verifier). Each figure contains three rows: (1) TTS performance vs. efficiency without verification on the test set, (2) the same analysis with verification, and (3) balanced accuracy on the validation set. The model with the highest balanced accuracy is selected as the cost-efficient choice and highlighted in rows 2 and 3.

Without verification, the largest model is usually the preferred choice due to its large accuracy advantage. With verification, there are cases where smaller models become compute-efficient choices because they achieve comparable TTS performance while requiring fewer FLOPs. This is consistent with our earlier observation in Section 5.1 RQ4. Notably, the generator selected by the highest balanced accuracy on the validation set (red circle) consistently matches the compute-efficient choice: it either identifies a smaller model that achieves accuracy close to the larger one or preserves the largest model when it retains a clear performance advantage. This shows that verification-based metrics provide effective guidance for generator selection.

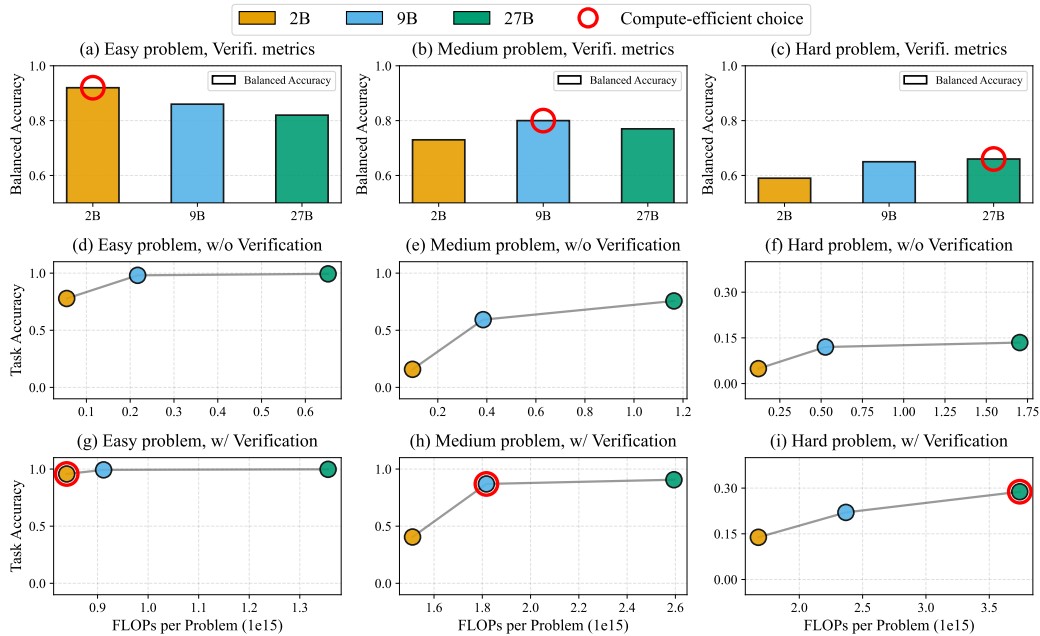

Figure 24: **Using a verification metric to select compute-efficient generators across different problem difficulties.** Qwen3-32B is used as the verifier, and generator candidates are Gemma2-2B, 9B, and 27B. (a–c) Balanced accuracy on the validation set varies across difficulty ranges and identifies different preferred generators: the 2B model on easy problems, 9B on medium problems, and 27B on hard problems. (d-f) TTS accuracy versus compute on the test set without verification. (g-i) TTS accuracy versus compute with verification. The generators selected by balanced accuracy correspond to the compute-efficient choices: on easier problems, smaller generators achieve accuracy close to larger models while requiring fewer FLOPs, whereas on hard problems, the largest generator retains a performance advantage.

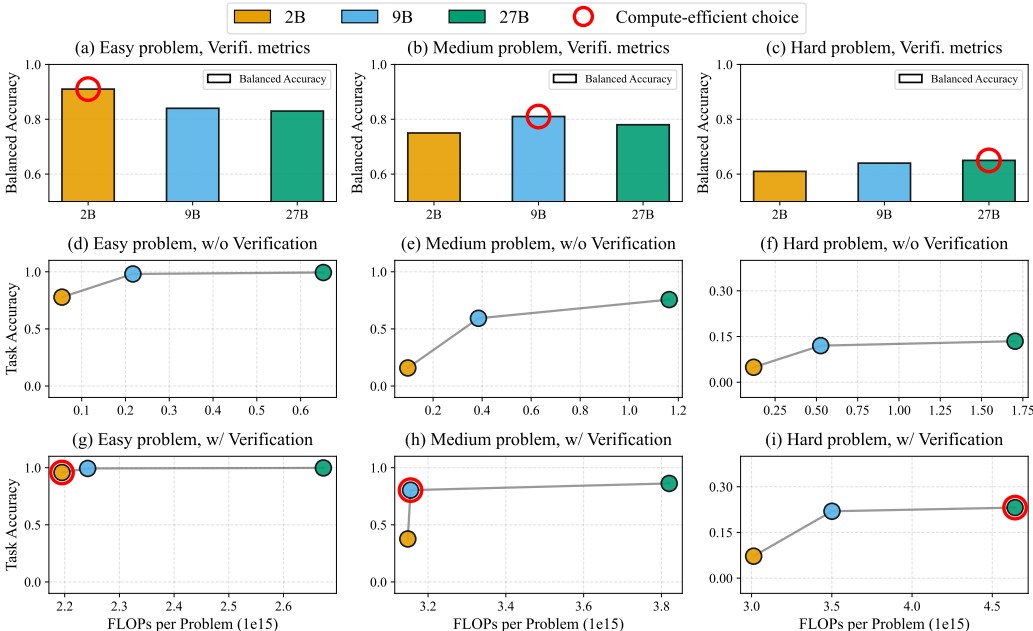

Figure 25: **Using a verification metric to select compute-efficient generators across different problem difficulties. GPT-4o is used as the verifier; task accuracy is measured by pass rate.** The experimental setups and observations are consistent with Figure 24.

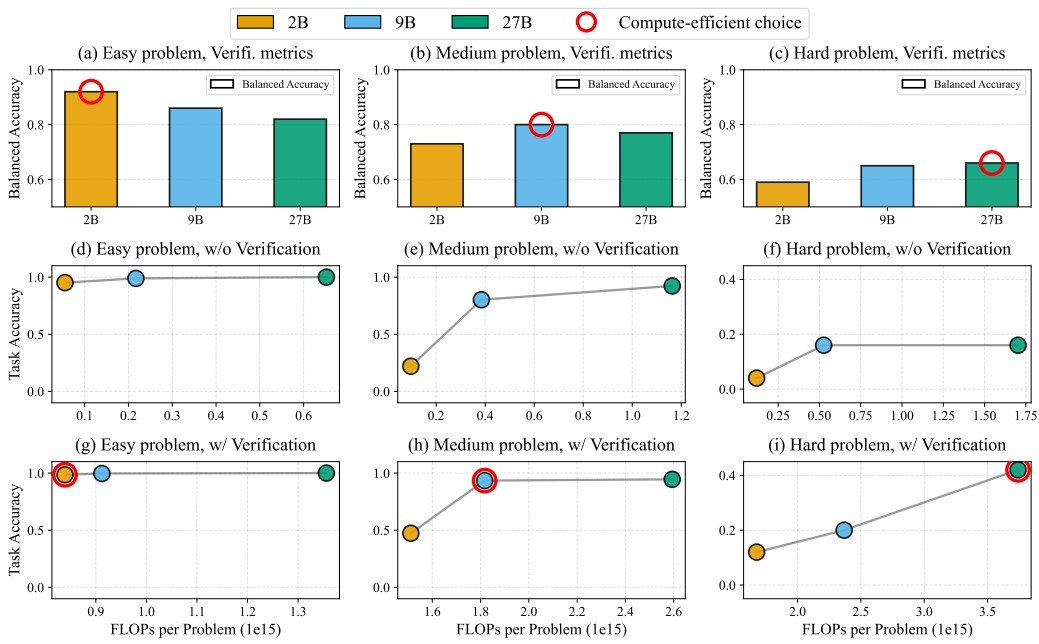

Figure 26: **Using a verification metric to select compute-efficient generators across different problem difficulties. Qwen3-32B is used as the verifier; task accuracy is measured by majority vote.** The experimental setups and observations are consistent with Figure 24.

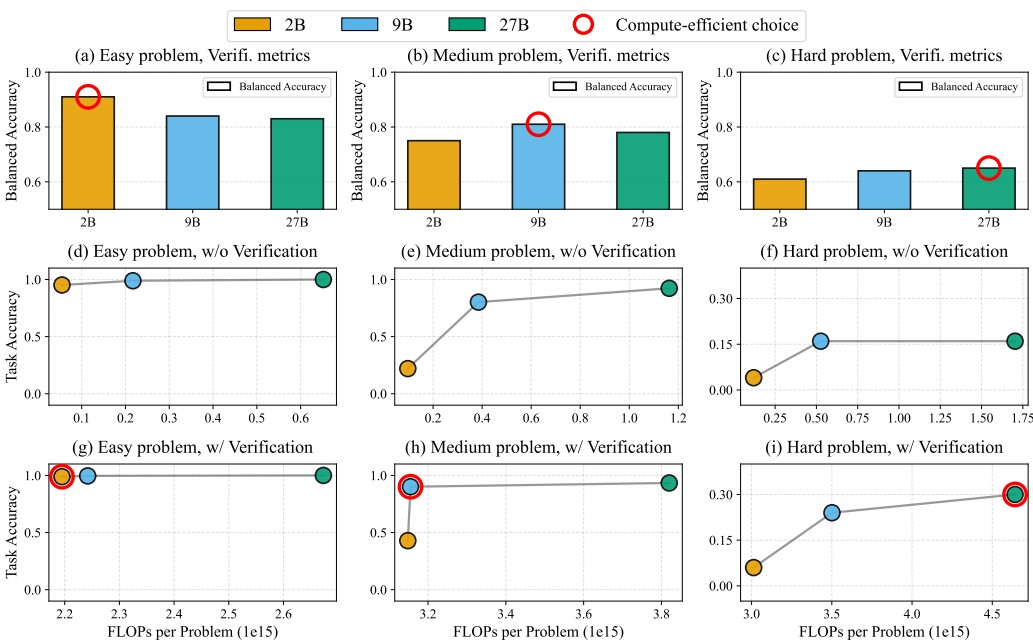

Figure 27: **Using a verification metric to select compute-efficient generators across different problem difficulties. GPT-4o is used as the verifier; task accuracy is measured by majority vote.** The experimental setups and observations are consistent with Figure 24.

### D.2.2 RQ5.1: HOW TO CHOOSE A VERIFIER GIVEN A FIXED GENERATOR?

We evaluate whether verification metrics can guide the choice between two verifiers (GPT-4o and Qwen2.5-7B) under different levels of problem difficulty. Following the procedure in Appendix D.2.1,

we compute the verification metric on the validation set and use it to select the verifier. TTS accuracy and computation cost are then reported on the test set.

**Results of RQ5.1** The results are presented in Figure 28 (pass rate) and Figure 29 (majority vote). For each figure, the first two subplots report TTS accuracy versus compute when using a small or large verifier. The final subplot reports the verification metrics.

On the hardest problem, GPT-4o yields only a small balanced accuracy improvement over Qwen2.5-7B. The minimal metric difference suggests the smaller verifier. Evaluated on the test set, the accuracy difference between the two verifiers is small. It confirms that the stronger verifier offers limited additional benefit in this regime. On hard problems, GPT-4o shows a clear gap of balanced accuracy over the small model. This difference points to choosing GPT-4o. On the test set, this leads to substantially improved TTS accuracy despite higher compute cost.

These results show that verification accuracy provides actionable guidance. When compute is limited, one can compare the balanced accuracy of the large and small verifiers: if the gap is small, the smaller verifier is the more cost-efficient choice; if the gap is large, the stronger verifier yields more substantial verification gains in TTS.

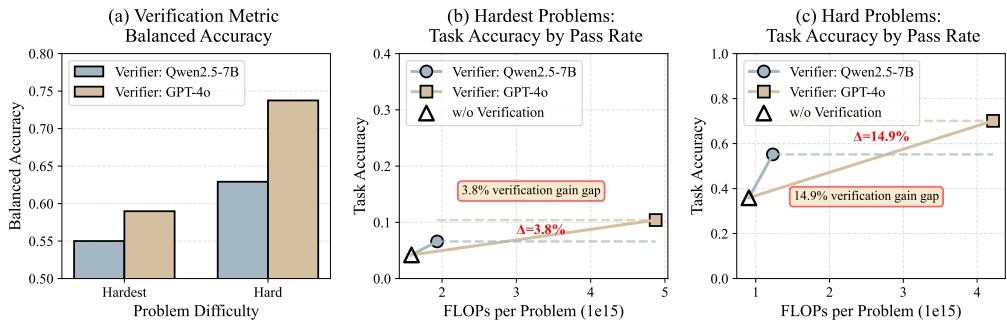

Figure 28: **Verification-metric–guided verifier selection across problem difficulty; task accuracy is measured by pass rate.** (a) Balanced accuracy on the validation set: when the metric gap between GPT-4o and Qwen2.5-7B is small (hardest problems), the smaller verifier is selected; when the gap is large (hard problems), the stronger verifier is selected. (b–c) Corresponding TTS accuracy on the test set: consistent with the metric-based choice, GPT-4o yields only a small improvement on the hardest problems but provides substantially larger gains on the hard problems.

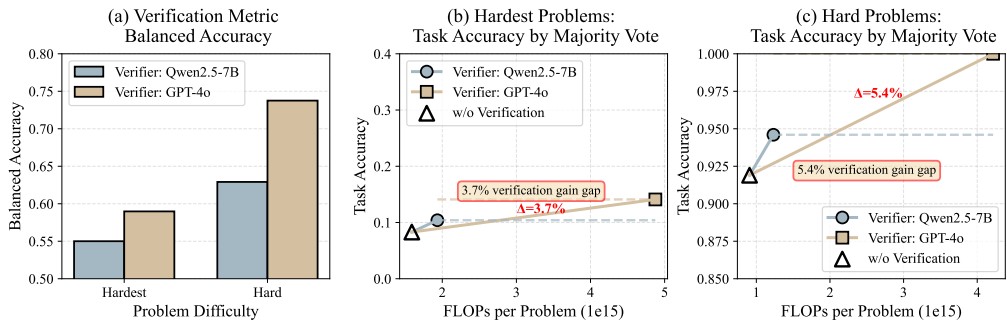

Figure 29: **Verification-metric–guided verifier selection across problem difficulty; task accuracy is measured by majority vote.** The experimental setups and observations are the same as Figure 28.

## D.3    ABLATION STUDY ON LARGE MODELS

We conduct an ablation study using a large LLM with more than 100B parameters, Qwen3-235B (Qwen/Qwen3-235B-A22B-Instruct-2507). We use the recommended sampling hyperparameters: temperature 0.7, top-p 0.8, and top-k 20. We repeat the experiments for RQ1 (Section 4.1) and RQ2

(Section 4.2) on the Mathematics domain and report the results in Figure 30. The results show that our main findings remain consistent when using a larger model.

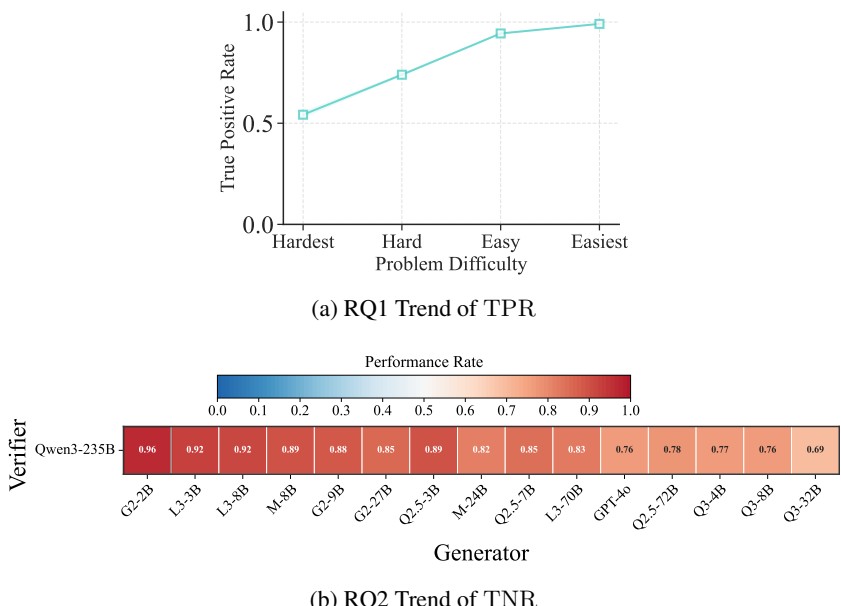

(a) RQ1 Trend of TPR

(b) RQ2 Trend of TNR

Figure 30: **Ablation study of our main findings with the large model Qwen3-235B.** The experiments follow the same setup as Sections 4.1 and 4.2 and are conducted on the Mathematics domain. (a) The trend that TPR increases as problem difficulty decreases is consistent with our RQ1 finding. (b) The trend that TNR decreases as the generator strength increases is consistent with our RQ2 finding.

### D.4 ABLATION STUDY ON VERIFICATION PROMPTS

We conduct an ablation study to evaluate the robustness of our findings under different verification prompts. All prompt templates used in this study are listed in Appendix A. In addition to the original "Verification Evaluation Prompt," we consider two alternatives: a concise "step-by-step" verification prompt from Zhang et al. (2024), and a "solve-then-verify" prompt that encourages the verifier to first solve the problem before evaluating the candidate response, a strategy shown to be effective in prior work Chen et al. (2025d). These two prompts differ substantially from our original design.

The experiments follow the same setup as Sections 4.1 and 4.2 and are conducted on the Mathematics domain. For each prompt, we apply Qwen2.5-72B as the verifier to evaluate responses produced by all 15 generator models. The results in Figure 31 show that our main findings regarding TPR and TNR dynamics remain consistent across all prompts, indicating that the observed verification dynamics are robust to prompt variations.

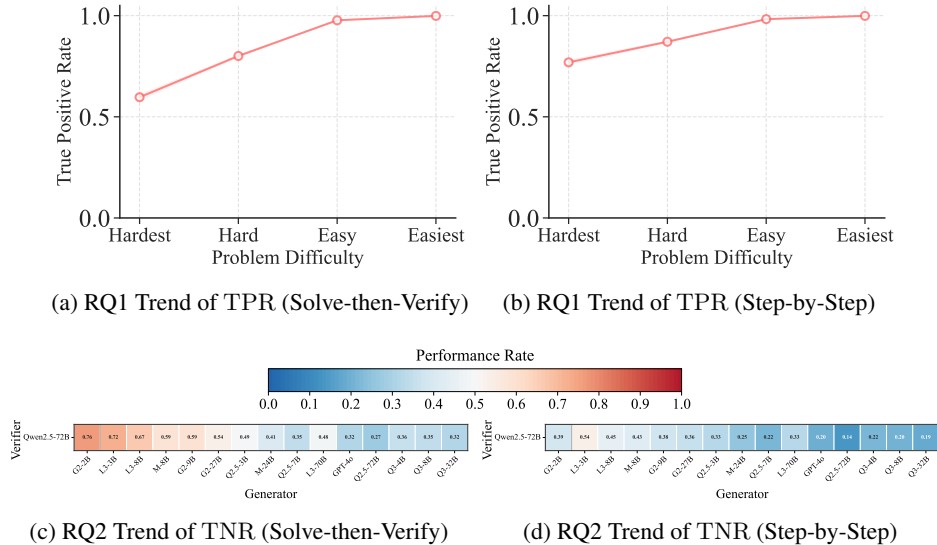

Figure 31: **Ablation study of our findings under different verification prompts.** (a, c) Using "solve-then-verify" prompt for verification. (b, d) Using "step-by-step" prompt for verification. The trend that TPR increases as difficulty decreases (RQ1) and TNR decreases as the generator strength increases (RQ2) still holds under both prompts.

Table 2: Comparison of TTS performance and token cost per problem when using a strong model as the verifier versus as the generator. Token count is reported on average per prompt.

| Generator | Verifier | Pass Rate | Avg. GPT-4o Tokens | Avg. Qwen3-4B Tokens |
|---|---|---|---|---|
| Qwen3-4B | – | 0.938 | – | 551 |
| GPT-4o | – | 0.952 | 482 | – |
| Qwen3-4B | GPT-4o | 0.954 | 193 | 551 |

## D.5 COMPARISON ANALYSIS OF STRONG MODEL AS GENERATOR OR VERIFIER

We examine whether it is practical to use a strong model as a verifier instead of using it solely as a generator. We compare two setups: (i) using GPT-4o as the generator, and (ii) using GPT-4o as the verifier together with a smaller generator, Qwen3-4B.

We use the same problem set as in Section 5.1: Mathematics problems with difficulty in the range $[0.7, 0.8)$, consisting of 181 problems. The TTS sampling size is 64. Task accuracy is measured using the pass rate, and efficiency is measured using the average tokens per prompt.

Here are the findings. Both setups achieve the same performance, but using GPT-4o as a verifier consumes 193 tokens on average per prompt, whereas GPT-4o as a generator uses 483 tokens per prompt on average. This represents a 2.5× reduction in token usage (and thus cost) per prompt. Thus, for practitioners who host small models but care about maximizing performance per dollar, leveraging the relatively short-form nature of verification with API models represents massive savings.

## E CASE STUDIES

> **Problem domain, dataset and difficulty**: Mathematics, MATH-500, Hardest ($d(x) = 0.075$)
> **Generator**: Qwen2.5-3B
> **Verifier**: GPT-4o
> **Question**: Bob and Alice each have a bag that contains one ball of each of the colors, blue, green, orange, red, and violet. Alice randomly selects one ball from her bag and puts it into Bob's bag. Bob then randomly selects one ball from his bag and puts it into Alice's bag. What is the probability that after this process the contents of the two bags are the same?
> **Ground-Truth Solution:** After Alice puts the ball into Bob's bag, his bag will contain six balls: two of one color and one of each of the other colors. After Bob selects a ball and places it into Alice's bag, the two bags will have the same contents if and only if Bob

picked one of the two balls in his bag that are the same color. Because there are six balls in the bag when Bob makes his selection, the probability of selecting one of the same colored pair is $2/6 = \boxed{\dfrac{1}{3}}$.

**Generator Output (Correct):** To determine the probability that the contents of both Bob's and Alice's bags are the same after the process described, we need to analyze each possible step and its impact on the contents of the bags.

Initially, each bag contains one ball of each of the colors: blue, green, orange, red, and violet.

### Step-by-Step Analysis [ *Details omitted* ]

### Identifying the Conditions for Equality [ *Details omitted* ]

### Calculating the Probability [ *Details omitted* ]

The probability is then calculated as follows:
$$\text{Probability} = \frac{\text{Number of favorable outcomes}}{\text{Total number of outcomes}} = \frac{2}{6} = \frac{1}{3}$$

Thus, the probability that after this process the contents of the two bags are the same is $\boxed{\dfrac{1}{3}}$.

**Verifier Output (False negative):** Explanation: The problem involves two steps: Alice transferring a ball to Bob's bag and then Bob transferring a ball back to Alice's bag. Initially, both bags have one ball of each color. After Alice transfers a ball, Bob's bag will have two balls of one color and one of each of the other four colors. For the bags to be identical after Bob's transfer, Bob must return the same color ball that Alice initially transferred. This is the only scenario where the bags will end up identical, as it restores the original distribution of colors in both bags.

The assistant's response correctly identifies that there are 6 possible outcomes for Bob's selection (since he has 6 balls to choose from) and that there are 2 favorable outcomes (returning one of the two balls of the color that Alice transferred). However, the assistant incorrectly concludes that the probability is 1/3. The correct probability should be 1/5, as Bob must specifically return the ball of the color that Alice transferred, which is only one specific ball out of the 6 in his bag. [Verifier generates a wrong answer 1/5 and uses it to compare with generator's correct answer 1/3]

Verdict: [[Incorrect]]

- - - - - - - - - - - - - - - - - - - - - - - - - - - - - - - - - - - - - - - - - - - - - - - - - -

**Problem domain, dataset and difficulty**: Natural Language Reasoning, FOLIO, Hardest ($d(x) = 0.122$)
**Generator**: Llama3.1-8B
**Verifier**: Qwen2.5-72B
**Question**: Some mammals have teeth. Platypuses have no teeth. Platypuses are mammals. Humans have teeth.

Based on the above, can the following be inferred? Platypuses are reptiles

**Ground-Truth Solution:** Uncertain

**Generator Output (Correct):** Reasoning: 1. The statement that some mammals have teeth and humans have teeth suggests that having teeth is a characteristic of mammals, but it's not exclusive to them. 2. Platypuses are indeed mammals, as stated in the information provided. 3. The fact that platypuses have no teeth does not contradict the fact that they are mammals, as having no teeth is also a characteristic of some mammals, as mentioned in point 1. 4. The only group of animals that are not mammals and have been ruled out as a category for platypuses are not the mammals, but rather the reptiles.

However, concluding that platypuses are reptiles because they have no teeth is not a sound inference, because having no teeth does not preclude an animal from being a mammal, as shown in point 3. Moreover, it contradicts the given statement that platypuses are indeed mammals.

5. Therefore, platypuses being reptiles cannot be inferred based on the given information.

Answer: Uncertain

**Verifier Output (False negative):** Explanation: The assistant's response is mostly correct in its reasoning but contains a minor inaccuracy in the final conclusion. The assistant correctly identifies that the fact that platypuses have no teeth does not make them reptiles, as some mammals also lack teeth. However, the assistant's final answer of "Uncertain" is not appropriate. Given the information provided, it is clear that platypuses are mammals, and there is no evidence to suggest they are reptiles. Therefore, the correct answer should be "No" because the statement "Platypuses are reptiles" can be definitively refuted based on the given information. [Although the verifier recognizes that the generator's reasoning process is correct, it rejects the conclusion due to a mismatch with its own (incorrect) answer.]

Verdict: [[Incorrect]]

Figure 32: **Case study: How problem difficulty impacts** TPR **(RQ1).** False negative verification on high-difficulty problems demonstrates why TPR decreases as problems become harder. When verifiers cannot correctly solve challenging problems, they produce incorrect reference solutions and reject generators' correct answers based on these flawed references.

**Problem domain, dataset and difficulty**: Mathematics, MATH-500, Easy ($d(x) = 0.571$)
**Generator**: Llama-3.1-8B
**Verifier**: Qwen2.5-72B
**Question**: Paul and Jesse each choose a number at random from the first six primes. What is the probability that the sum of the numbers they choose is even?

**Ground-Truth Solution:** [ *Details omitted* ] Therefore, the probability that the sum of the numbers Paul and Jesse choose IS even is $1 - \frac{5}{18} = \boxed{\dfrac{13}{18}}$.

**Generator Output (Incorrect):** To find the probability that the sum of the numbers Paul and Jesse choose is even, we need to first understand what numbers can make an even sum.

The sum of two numbers is even if both numbers are either even or odd.

Since the first six primes are 2, 3, 5, 7, 11, and 13, we have only one even prime, which is 2.

Now, let's consider the possible scenarios:

1. Both Paul and Jesse choose the number 2. In this case, the sum is even.

2. One of Paul or Jesse chooses the number 2, and the other chooses an odd prime. In this case, the sum is odd.

3. Both Paul and Jesse choose odd primes. In this case, the sum is even. [Generator correctly identifies two cases for even sum, one case for odd sum.]

[ *Probability calculations omitted omitted* ]

Now, let's add the probabilities of the first two scenarios (both choose 2, and one chooses 2 and the other an odd prime) and the third scenario (both choose an odd prime) to get the total probability of an even sum:

Probability = 1/36 + 10/36 + 25/36 Probability = 36/36 Probability = 1

The probability that the sum of the numbers Paul and Jesse choose is even is 1 or 100%. This is because it is impossible for the sum to be odd, as they will always choose either two even numbers or two odd numbers. [Generator makes self-contradiction: Earlier stated scenario 2 gives odd sum, now count scenario 2 as even sum and claims odd sum is impossible]

**Verifier Output (True negative):** The assistant's response contains a logical error in the final calculation. The assistant correctly identifies the scenarios where the sum of the numbers chosen by Paul and Jesse is even. However, the assistant incorrectly adds the probabilities of all scenarios, including the scenario where the sum is odd, which should not be included in the final probability calculation. [Verifier catches the self-contradiction of generator.]

The correct calculation should only include the probabilities of the scenarios where the sum is even: - Both choose 2: $\frac{1}{36}$ - Both choose odd primes: $\frac{25}{36}$

The probability of the sum being even is the sum of these two probabilities:

$$\frac{1}{36} + \frac{25}{36} = \frac{26}{36} = \frac{13}{18}$$

Therefore, the final answer should be $\boxed{\frac{13}{18}}$. [Verifier can solve the problem correctly]

Verdict: [[Incorrect]]

- - - - - - - - - - - - - - - - - - - - - - - - - - - - - - - - - - - - - - - - - - - - - - - - - - - - - -

**Same problem and verifier as above with a strong generator**:
**Generator**: GPT-4o
**Verifier**: Qwen2.5-72B
**Generator Output (Incorrect):** To find the probability that the sum of the numbers chosen by Paul and Jesse from the first six prime numbers is even, we proceed as follows:

1. **List the first six prime numbers**: The first six prime numbers are 2, 3, 5, 7, 11, and 13.

2. **Identify even and odd primes**:

- Even prime: 2

- Odd primes: 3, 5, 7, 11, 13

3. **Understand when the sum is even**:

- A sum is even if both numbers are even or both numbers are odd.

- Since there is only one even prime (2), for the sum to be even, both numbers must be odd. [Fatal error: Ignores that both choosing 2 (even+even=even) is also possible.]

4. **Calculate the total number of ways to choose numbers**: [ *Calculation details omitted omitted* ]

5. **Calculate the number of ways to get an even sum**: [ *Calculation details omitted omitted* ]

6. **Calculate the probability**: [ *Calculation details omitted omitted* ]

Thus, the probability that the sum of the numbers they choose is even is $\boxed{\frac{25}{36}}$. [Final answer is wrong because following steps consistently based on fatal error made above.]

**Verifier Output (False positive):** Explanation: The assistant's response is logically sound and mathematically accurate. Here's a breakdown of the steps:

1. **Identifying the first six primes**: The assistant correctly identifies the first six prime numbers as 2, 3, 5, 7, 11, and 13.

2. **Condition for the sum to be even**: The assistant correctly states that for the sum to be even, both numbers must be odd (since the only even prime is 2, and choosing 2 would make the sum odd if the other number is odd). [Verifier is convinced by the generator and ignores the error here.]

3. **Total number of possible outcomes**: The assistant correctly calculates the total number of outcomes as 6 * 6 = 36.

4. **Number of outcomes where both numbers are odd**: The assistant correctly identifies that there are 5 odd primes and calculates the number of outcomes where both numbers are odd as 5 * 5 = 25.

5. **Probability calculation**: The assistant correctly calculates the probability as the number of favorable outcomes (25) divided by the total number of outcomes (36), resulting in $\frac{25}{36}$.

The assistant's reasoning and calculations are correct throughout the response. [Although the verifier demonstrated correct problem-solving when evaluating Llama3.1-8B in last example, it doesn't apply the same mechanism to GPT-4o, instead accepting the GPT-4o's incorrect reasoning.]

Verdict: [[Correct]]

Figure 33: **Case study: How generator capability influences** TNR **(RQ2).** We compare verifier behavior on the same problem when evaluating weak (Llama3.1-8B) versus strong (GPT-4o) generators. The weak generator correctly lists all the cases but then contradicts itself in the final calculation, producing an inconsistent solution. The verifier catches this apparent mistake and rejects it. The strong generator makes a single early-stage error but maintains internal consistency throughout its reasoning, generating a well-structured and self-contained chain that appears correct. Despite the solution containing an actual error, the verifier fails to detect it and accepts the solution.

