# OpenReview forum: "Variation in Verification: Understanding Verification Dynamics in Large Language Models"
_ICLR.cc/2026/Conference — ICLR 2026 Poster_

### Official Review · Reviewer_1wQz · 2025-10-16

**Soundness:** 3
**Presentation:** 3
**Contribution:** 3
**Rating:** 8
**Confidence:** 4

**Summary:**

This paper investigates the factors that influence verification performance when large language models (LLMs) are used to assess the correctness of solutions generated by other LLMs. The authors conduct a systematic empirical study across three dimensions: problem difficulty, generator capability, and verifier generation capability. Using 12 benchmarks spanning mathematical reasoning, knowledge QA, and natural language reasoning, they evaluate 14 open-source models and GPT-4o in both generator and verifier roles. The study reveals three key findings: (1) verifiers more reliably certify correct responses on easier problems, (2) weak generators produce errors that are easier to detect than those from strong generators, and (3) verification ability correlates with verifier generation capability in a difficulty-dependent manner. The authors demonstrate practical implications for test-time scaling, showing that weak generators can nearly match strong generators when paired with the same verifier, and identifying regimes where strong verifiers provide limited advantages over weak ones.

**Strengths:**

- The paper addresses an important and timely problem, as LLM-based verification is increasingly central to test-time scaling methods and the efficient deployment of language models in production systems.
- The experimental design is comprehensive and well-controlled, systematically varying problem difficulty, generator capability, and verifier capability across 12 benchmarks and 15 models to isolate the effect of each factor.
- The finding that generator capability primarily influences error detectability (TNR) rather than correct response recognition (TPR) is novel and provides actionable insights for model selection in verification pipelines.
- The case studies in Figures 20 and 21 effectively illustrate the mechanisms behind the main findings, showing how problem difficulty affects TPR through reference solution quality and how generator capability affects TNR through solution consistency.
- The practical implications for test-time scaling are clearly demonstrated, showing substantial performance gap reductions between weak and strong generators when using the same verifier (Section 5.1 and Figure 17).
- The paper identifies regimes where strong verifiers provide minimal additional benefit over weak verifiers, which has important cost-efficiency implications for practitioners deploying verification systems at scale.
- The difficulty-dependent correlation analysis in Section 4.3 reveals nuanced patterns (saturated, linear, threshold-limited) that prior work assuming simple linear relationships overlooked.

**Weaknesses:**

- The paper lacks theoretical analysis or mechanistic explanations for the observed patterns, relying entirely on empirical observations without providing deeper understanding of why these verification dynamics emerge.
- The definition of problem difficulty as average pass rate across generators is model-dependent and may not capture intrinsic problem characteristics, potentially confounding difficulty effects with model-specific biases.
- The experimental scope is limited to verifiable problems with objective ground-truth answers, which excludes important domains like open-ended generation, creative writing, and subjective evaluation tasks where verification is also needed.
- The paper does not address the computational costs of verification, which is a critical practical consideration given that verification gains may not justify the additional compute in all scenarios.
- The analysis of reasoning models (Appendix C.2) is relegated to the appendix despite showing different TNR behavior, suggesting that the main findings may not fully generalize to this increasingly important class of models.
- The selection of specific prompt templates, sampling hyperparameters, and verification evaluation procedures is not thoroughly ablated, leaving uncertainty about the robustness of findings to these design choices.
- The paper identifies regimes where verification provides limited gains but does not propose actionable strategies for overcoming these fundamental limitations beyond avoiding those regimes.
- Some experimental details are missing or unclear, such as the exact procedure for handling invalid verifier outputs, the rationale for the specific 4 correct/4 incorrect sampling strategy, and how the fallback LLM-based correctness checking affects results.

**Questions:**

1. How sensitive are the main findings to the choice of prompt templates for generation and verification? Did you experiment with alternative prompting strategies, and if so, how did they affect the observed patterns?
2. In Section 4.1, you show that problem difficulty primarily affects TPR. However, for reasoning models in Appendix C.2, TNR also increases with easier problems. Can you provide more insight into why extended reasoning changes this relationship?
3. The definition of problem difficulty as average pass rate across generators may be influenced by the specific set of models used. How would the findings change if you used a different set of generators or defined difficulty based on human performance?
4. For the TTS experiments in Section 5, you use K=64 samples per problem. How sensitive are the verification gains to this choice? Is there a point of diminishing returns where additional samples provide minimal benefit?
5. The paper identifies regimes where strong verifiers provide limited additional benefit (Section 5.2). Are there alternative verification strategies (e.g., multi-verifier consensus, uncertainty-aware verification) that might overcome these limitations?

---

> ### Author Response · Authors · 2025-11-21
> **Response to Reviewer 1wQz (Part 1)**
>
> We thank the reviewer for their thorough review. In particular, we are delighted you found our work "timely" and "comprehensive" with "practical implications". Here, we respond to comments and questions point-by-point.
>
> &nbsp;
>
> > Weakness 1: The paper lacks theoretical analysis or mechanistic explanations for the observed patterns, relying entirely on empirical observations without providing a deeper understanding of why these verification dynamics emerge.
>
> We agree that a more systematic analysis of observed patterns would shed further insights into our findings. While we included case studies for both RQ1 and RQ2, here we conduct a more systematic analysis. The complete explanation has been updated in L282 and L356, Figures 3 and 5 (large-scale analysis), Figures 32 and 33 (case studies).
>
> For RQ1, our main finding was "easy problems allow verifiers to more reliably certify correct responses". Our case study (L279, Figure 32) showed that verifiers tend to generate a reference answer to anchor verification. As problem difficulty increases, the verifier is unable to generate accurate references, leading to more false negatives. We conduct a global meta-analysis by using GPT-4.1-mini to determine if (1) a verifier is attempting to generate a reference and, if yes, (2) if the generated reference contains a mistake. For (2), we include the ground-truth solution (i.e., reference-based meta-analysis). We analyze 367,920 verification traces from GPT-4o, which span 2,347 math problems across 15 generators. As we show in L282, Figure 3, as problem difficulty increases, the verifier increasingly makes mistakes in generating the reference answer, with 39.1% of verification false negatives containing a reference answer mistake in the hard set of problems. In contrast, on easy problems, there are substantially fewer false negatives and reference answer-related mistakes, showing that faulty reference generation is a large driving factor of false negatives.
>
> For RQ2, we found that “weak generators produce errors that are easier to detect than strong generators," where our case study showed that weaker generated produced extremely obvious, surface-level errors (e.g., self-contradictions). Here, we prompt GPT-4.1-mini to determine if a generator's response contains a surface-level error (self-contradictions, basic arithmetic or factual mistakes, or incomplete generations). As we show in L356, Figure 5, analysis over 98,782 generator responses (the same ones used in S4.2 for RQ2) finds a direct correlation between generator strength and surface-level error frequency. This confirms that our case study presents a systematic mechanism of response quality.
>
> &nbsp;
>
> > Weakness 2: The definition of problem difficulty as average pass rate across generators is model-dependent and may not capture intrinsic problem characteristics, potentially confounding difficulty effects with model-specific biases.
>
> We estimate the difficulty of each problem by averaging the pass rate of 15 models, ranging from 2B and 72B, across 4 model families, and proprietary model GPT-4o. The set of models for difficulty estimation fully covers the set of models used for all analysis studies. This design is intended to minimize the model-specific biases by averaging over a diverse pool of models of varying strengths; Similar approaches for problem difficulty estimation have been adopted in other studies [1].
>
> &nbsp;
>
> > Question 3: The definition of problem difficulty as average pass rate across generators may be influenced by the specific set of models used. How would the findings change if you used a different set of generators or defined difficulty based on human performance?
>
> **Our findings hold even when difficulty is estimated using one generator model and practical label-free estimators; See Appendix D.1, Figures 22 and 23.** We conducted additional experiments evaluating our difficulty-related analyses in RQ1 and RQ3 using a practical uncertainty-based estimator.
>
> We adopt the uncertainty-based estimator [2]. For each question, we draw 8 completions from Qwen2.5-3B-instruct, extract the final answers, compute the empirical answer distribution, and compute the normalized entropy in [0, 1]. A score close to 1 indicates that all 8 answers differ, corresponding to high model uncertainty and therefore high estimated difficulty.
>
> Implementation details are reported in Appendix D.1. The trends are consistent with those in Figures 2 and 6 that use oracle difficulty, demonstrating that our main conclusions about verification dynamics continue to hold even under an approximate, uncertainty-based difficulty estimator.
>
> &nbsp;
>
> **Reference**
>
> [1] Snell et al. Scaling LLM Test-Time Compute Optimally can be More Effective than Scaling Model Parameters
>
> [2] Lee et al. Semantic exploration with adaptive gating for efficient problem solving with language models.

---

> ### Author Response · Authors · 2025-11-21
> **Response to Reviewer 1wQz (Part 2)**
>
> > Weakness 3: The experimental scope is limited to verifiable problems with objective ground-truth answers, which excludes important domains like open-ended generation, creative writing, and subjective evaluation tasks where verification is also needed.
>
> We intentionally choose to focus on verifiable problems to enable controlled experimental design: Verifiable answers serve as a valuable, objective measure of both verifier and generator performance, allowing us to compute things like problem difficulty, verifier accuracy, and generator capability at scale. Note that we deploy verifiers in *reference-free* manner, mimicing how verifiers would fare in open-ended evaluation settings. We lay out this reasoning in L92.
>
> &nbsp;
>
> > Weakness 4: The paper does not address the computational costs of verification, which is a critical practical consideration given that verification gains may not justify the additional compute in all scenarios.
>
> We provide additional study on analyzing computational costs of verification. These results show how our findings help answer two questions:
>
> 1. When is it worth using a compute-heavy verifier? In Appendix D.2.2, we show that our findings in RQ5 offer guidance on when the cost of a strong verifier is worthwhile. The gap between small and large verifiers varies with problem difficulty and can be predicted by verification metrics (balanced accuracy). This allows us to identify regimes where a strong verifier (e.g., GPT-4o) produces substantial TTS gains and regimes where the improvement is small relative to its compute cost. For difficult problems, GPT-4o provides only a small gain despite requiring more inference compute, whereas for medium-difficulty problems the stronger verifier yields larger improvements.
> 2. Is it more compute-efficient to use a strong model as a verifier or as a generator? In Appendix D.5, we compare using a strong model for generation versus for verification. We find that using a small generator (Qwen3-4B) with a strong verifier (GPT-4o) achieves essentially the same TTS accuracy as using GPT-4o directly as the generator, while reducing GPT-4o token consumption by 2.5x per prompt. This shows that verification can be more compute-efficient than generation, and that using a strong model as a verifier can be more cost-effective than using it as a generator.
>
> &nbsp;
>
> > Weakness 5: The analysis of reasoning models (Appendix C.2) is relegated to the appendix despite showing different TNR behavior, suggesting that the main findings may not fully generalize to this increasingly important class of models.
>
> We do not believe there is a fundamental mismatch between results presented in Figure 11(d-f) and those presented in Figure 2, as *our findings for RQ1,  which form the basis of Figure 2, do not rely at all on TNR*: We draw all conclusions from TPR, for which reasoning models and instruction tuned models exhibit the same TPR behavior as a function of problem difficulty. Regarding TNR vs. problem difficulty, Figure 2 shows that for some instruction-tuned models, TNR increases vs. problem difficulty, whereas for others it decreases; Overall, the impact of problem difficulty on TNR is extremely mixed, varying on a model-to-model basis. Moreover, our results for TNR, the focus on RQ2, also hold with reasoning models, as shown in Figure 13 and supporting discussion in Appendix C.2.
>
> &nbsp;
>
> > Question 2: In Section 4.1, you show that problem difficulty primarily affects TPR. However, for reasoning models in Appendix C.2, TNR also increases with easier problems. Can you provide more insight into why extended reasoning changes this relationship?
>
> On easy problems, strong generators can still produce incorrect responses that are internally consistent and therefore difficult for standard instruction-tuned verifiers to detect. These errors are typically subtle: the generator makes a small early mistake but then produces a coherent and well-structured chain of reasoning that appears correct. Current large instruction models are often unable to identify such errors, which limits TNR on easy problems shown in Section 4.1. Reasoning models, however, are better at identifying these subtle but coherent mistakes. This can increase TNR on easy problems, because extended-reasoning verifiers are capable of performing deeper checking and reflection.
>
> We support this explanation with the case study in Appendix E (Figure 33). In this example, GPT-4o produces an incorrect solution to an easy math question. The error occurs early, but the rest of the chain is fully consistent and well-structured. Qwen2.5-72B fails to detect this error, which may help explain why TNR does not improve with lower difficulty for standard instruction models. An extended reasoning verifier would likely be able to detect the error, explaining why TNR increases for such models in Appendix C.2.

---

> ### Author Response · Authors · 2025-11-21
> **Response to Reviewer 1wQz (Part 3)**
>
> > Question 6: For the TTS experiments in Section 5, you use K=64 samples per problem. How sensitive are the verification gains to this choice? Is there a point of diminishing returns where additional samples provide minimal benefit?
>
> As defined in Equation (1), our TTS performance metric $\hat{p}_{G, V}(\mathcal{D} ; K)$ is invariant to $K$ in expectation. It measures the expected accuracy obtained by uniformly sampling a single response from the pool of verifier-retained responses. This expectation-based view captures the average quality of verifier-retained responses without tying performance to a specific selection strategy. Because this metric is an expectation over a uniform distribution, increasing $K$ does not change its mean value; it only reduces the variance of the estimate. In practice, we found that $K=64$ is sufficiently large to yield stable accuracy estimates with low variance.
>
> &nbsp;
>
> > Weakness 6: The selection of specific prompt templates, sampling hyperparameters, and verification evaluation procedures is not thoroughly ablated, leaving uncertainty about the robustness of findings to these design choices.
>
> > Question 1: How sensitive are the main findings to the choice of prompt templates for generation and verification? Did you experiment with alternative prompting strategies, and if so, how did they affect the observed patterns?
>
> For the generation prompt, we used five prompts, each for one different type of task. Our findings across different tasks are consistent, which demonstrates robustness over generation prompts. For the verification prompt, we demonstrate robustness across different prompt templates by conducting experiments (Appendix D.4, Figure 31) using two additional verification prompts, which are updated in Appendix A. One prompt is from [1] and requests "step-by-step" verification, which is more concise than the original prompt. The other prompt encourages the verifier to perform "solve-then-verify," a strategy shown to be beneficial for evaluation in prior work [2].  The results show that our main findings regarding the TPR and TNR dynamics remain consistent under these alternative prompting strategies.
>
> For sampling hyperparameters, we use the hyperparameters suggested by the model documentation to ensure they achieve expected performance.
>
> &nbsp;
>
> > Weakness 7: The paper identifies regimes where verification provides limited gains but does not propose actionable strategies for overcoming these fundamental limitations beyond avoiding those regimes.
>
> > Question 5: The paper identifies regimes where strong verifiers provide limited additional benefit (Section 5.2). Are there alternative verification strategies (e.g., multi-verifier consensus, uncertainty-aware verification) that might overcome these limitations?
>
> We clarify that our goal is to analyze the factors that influence verification performance and to characterize the resulting verification dynamics. The regimes where verification provides limited gains naturally arise from these dynamics. Our analysis already provides practical guidance: in Appendix D.2.2, we show how these regimes can be used to choose compute efficient strategies, such as determining when the additional cost of a strong verifier leads to meaningful improvements and when it does not.
>
> Exploring alternative verification strategies, including using multiple verifiers or incorporating uncertainty information, is an exciting avenue of future work. Our work is meant to guide future methodological efforts by giving researchers an understanding of the fundamental phenomenon that influence verification; Armed with such insights, we hope to produce more reliable verification systems.
>
> &nbsp;
>
> **Reference**
>
> [1] Zhang et al. Generative Verifiers: Reward Modeling as Next-Token Prediction
>
> [2] Chen et al. RM-R1: Reward Modeling as Reasoning

---

> > ### Comment · Reviewer_1wQz · 2025-11-27
> > **Response to Authors**
> >
> > Thank you for your comprehensive response! After reading the rebuttal, I will maintain my positive score.

---

> > > ### Author Response · Authors · 2025-11-27
> > > **Reply Rebuttal Comment by Authors**
> > >
> > > We thank the reviewer for reading our rebuttal and for the positive score. Thanks again for the constructive feedback.

---

### Official Review · Reviewer_QrzN · 2025-11-01

**Soundness:** 2
**Presentation:** 2
**Contribution:** 2
**Rating:** 4
**Confidence:** 3

**Summary:**

This paper presents a empirical study on the dynamics of LLM-based verification for test-time scaling. The authors investigate how problem difficulty, generator capability, and verifier generation capability influence verification performance across mathematical reasoning, knowledge, and natural language reasoning tasks. Key findings include that easy problems allow verifiers to certify correct responses more reliably, errors from weaker generators are easier to detect, and verifier capability correlates with performance in a difficulty-dependent manner that varies by problem difficulty.

**Strengths:**

1. The paper conducts a thorough investigation across three crucial dimensions (problem difficulty, generator capability, verifier capability) and evaluates performance on a wide array of 12 benchmarks using 14 diverse LLMs.
2. The paper details its experimental setup, metrics, model and prompts, ensuring high reproducibility.
3. The findings provide concrete guidance for optimizing TTS applications, such as identifying scenarios where weaker generators can achieve performance comparable to stronger ones.

**Weaknesses:**

1.  Despite conducting extensive analytical experiments, many conclusions remain relatively intuitive rather than providing deeper theoretical insights. For instance, the findings that "easy problems allow verifiers to more reliably certify correct responses" and "weak generators produce errors that are easier to detect than strong generators" align with common intuition but lack mechanistic explanations about why these patterns emerge.
2. The use of pass rate as the sole difficulty metric (d(x)) may oversimplify the complex nature of problem difficulty, potentially missing important nuances in what makes problems challenging for verification.

**Questions:**

1. How might your findings apply to other verifiable domains, such as coding?
2. The paper highlights distinct non-linear correlation patterns between verifier generation capability and verification performance (e.g., saturated, linear, threshold-limited). Could the authors offer a more in-depth theoretical or empirical analysis of why these different regimes emerge across varying problem difficulties?
3. Did you explore different verification prompting strategies? How might alternative verification frameworks affect the observed dynamics?

---

> ### Author Response · Authors · 2025-11-21
> **Response to Reviewer QrzN (Part 1)**
>
> We thank the reviewer for their review and are glad that you found our work "thorough" and one that "provide[s] concrete guidance". We respond to critiques and comments below.
>
> &nbsp;
>
> > Weakness 1: Despite conducting extensive analytical experiments, many conclusions remain relatively intuitive rather than providing deeper theoretical insights. For instance, the findings that "easy problems allow verifiers to more reliably certify correct responses" and "weak generators produce errors that are easier to detect than strong generators" align with common intuition but lack mechanistic explanations about why these patterns emerge.
>
> We agree that mechanistic explanations of observed patterns would shed further insights into our findings. While we included case studies for both RQ1 and RQ2 findings (the two mentioned by the reviewer), we conducted a more systematic analysis. The complete explanation has been updated in L282 and L356, Figures 3 and 5 (large-scale analysis), Figures 32 and 33 (case studies).
>
> For RQ1, our main finding was "easy problems allow verifiers to more reliably certify correct responses". Our case study (Figure 32) showed that verifiers tend to generate a reference answer to anchor verification. As problem difficulty increases, the verifier is unable to generate accurate references, leading to more false negatives. We conduct a global meta-analysis by using GPT-4.1-mini to determine if (1) a verifier is attempting to generate a reference and, if yes, (2) if the generated reference contains a mistake. For (2), we include the ground-truth solution (i.e., reference-based meta-analysis). We analyze 367,920 verification traces from GPT-4o, which span 2,347 math problems across 15 generators. As we show in L282, Figure 3, as problem difficulty increases, the verifier increasingly makes mistakes in generating the reference answer, with 39.1% of verification false negatives containing a reference answer mistake in the hard set of problems. In contrast, on easy problems, there are substantially fewer false negatives and reference answer-related mistakes, showing that faulty reference generation is a large driving factor of false negatives.
>
> For RQ2, we found that "weak generators produce errors that are easier to detect than strong generators", where our case study (Figure 33) showed that weaker generators produced extremely obvious, surface-level errors (e.g., self-contradictions). Here, we prompt GPT-4.1-mini to determine if a generator's response contains a surface-level error (self-contradictions, basic arithmetic or factual mistakes, or incomplete generations). As we show in L356, Figure 5, analysis over 98,782 generator responses (the same ones used in S4.2 for RQ2) finds a direct correlation between generator strength and surface-level error frequency. This confirms that our case study presents a systematic mechanism of response quality.
>
> &nbsp;
>
> > Weakness 2: The use of pass rate as the sole difficulty metric (d(x)) may oversimplify the complex nature of problem difficulty, potentially missing important nuances in what makes problems challenging for verification.
>
> **Clarifying concerns about the difficulty metric.** The reviewer notes that using pass rate as a difficulty metric may oversimplify the nature of problem difficulty. We use problem difficulty only as an independent variable to study how it affects verification performance, not as the explanation for the observed verification behavior. Our study aims to understand how different factors, including problem difficulty, influence verification performance. Following prior work [1], we use problem generation difficulty, which reflects how hard the problem is to solve and is a standard property studied in understanding LLM behavior and capability. After observing and reporting how verification performance varies with this notion of problem difficulty in S4.1 L320, we then examine the mechanism behind this relationship in L356. The explanation of the nuances is presented in the last response.
>
> &nbsp;
>
> > Question 1: How might your findings apply to other verifiable domains, such as coding?
>
> We study three verifiable domains in the paper, including mathematics, natural language reasoning, and knowledge, and our main findings are consistent across these domains. The 15 models we analyze are general-purpose instruction models with varying mathematical, reasoning, and coding abilities. We do not expect substantial differences for coding tasks. Extending the analysis to coding is a natural next step, and we will include this in future work.
>
> &nbsp;
>
> **Reference**
>
> [1] Snell et al. Scaling LLM Test-Time Compute Optimally can be More Effective than Scaling Model Parameters

---

> ### Author Response · Authors · 2025-11-21
> **Response to Reviewer QrzN (Part 2)**
>
> > Question 2: The paper highlights distinct non-linear correlation patterns between verifier generation capability and verification performance (e.g., saturated, linear, threshold-limited). Could the authors offer a more in-depth theoretical or empirical analysis of why these different regimes emerge across varying problem difficulties?
>
> In addition to Figure 6, the empirical results showing the non-linear correlation patterns appear in Appendix C.4 (Figure 14), where we identify the three regimes across all difficulty ranges. Below, we provide a clearer analysis of these patterns, and it has been updated in Appendix C.4.
>
> **Threshold-limited pattern on difficult problems.** The threshold-limited regime appears in difficult problems, for example, in Figure 14 (a) in the difficulty range $[0.2, 0.3)$. In this regime, increasing the verifier’s generation capability does not translate into noticeably better verification, which is observed as the saturation pattern. As an example, Qwen3-32B (largest rectangle) has a generation capability of about 0.6, yet its verification performance is nearly identical to Qwen2.5-72B (circle) with a lower generation capability of around 0.4. Both models plateau around a verification score of 0.7.
>
> The underlying reason is that on extremely difficult problems, even the verifiers with the best generation capability are not that good (<=0.6 pass rate). As a result, pushing generation ability doesn't yield many gains, as the models still lack the intrinsic ability to "handle" these problems, whether it be solving or verifying. This implies that reliable verification requires the verifier to reach a sufficiently high level of generation capability. When models fall below this requirement on difficult problems, verification performance plateaus.
>
> **Transition to a linear pattern on medium problems.** When moving to medium-difficulty problems, such as those in Figure 14 (a) range $[0.4, 0.5)$, the relationship becomes more linear. In this regime, Qwen3-32B reaches a higher generation capability (around 0.8), and its verification performance increases accordingly to 0.75. This improvement allows Qwen3-32B to outperform another model with a generation capability of around 0.5 and verification around 0.7, indicating a linear pattern.
>
> In this intermediate regime, we see the most linear correlation. These problems are of average difficulty, and models typically demonstrate a baseline level of ability in solving such problems, as measured by generation capability. As such, we find that this regime matches well with regimes analyzed in prior work [1,2]. Therefore, this region exhibits the most intuitive behavior: Model verification ability tracks directly with its ability to understand (i.e., solve) the problem. It's important to note that these problems are **solvable** (i.e., not extremely difficult).
>
> **Saturated pattern on easy problems.** In Figure 14(a) for the difficulty range $[0.9, 1.0)$, Qwen2.5-7B (medium circle) already reaches nearly 1.0 in generation capability, the same as the stronger Qwen3-32B. Their difference on the $x$-axis is therefore essentially zero, yet their verification performance differs by about 0.1. This illustrates the saturated pattern.
>
> The underlying reason is that differences in verification performance come from factors other than the ability to solve the problem itself. One such factor is whether the verifier can detect more subtle or advanced mistakes made by strong generators. As illustrated in our case studies (Figure 33), a model may be able to solve a problem correctly but still fail to recognize certain non-obvious or high-level errors in another model’s reasoning. Thus, perfect generation capability is not sufficient for achieving strong verification performance on easy problems.
>
> &nbsp;
>
> > Question 3: Did you explore different verification prompting strategies? How might alternative verification frameworks affect the observed dynamics?
>
> Yes. We conducted additional experiments (Appendix D.4, Figure 31) using two verification prompts that differ from the one used in the main draft. The results show that our main findings regarding the TPR and TNR dynamics remain consistent under these alternative prompting strategies. One prompt is from [3] and requests "step-by-step" verification, which is more concise than our original prompt. The other prompt encourages the verifier to perform "solve-then-verify," a strategy shown to be beneficial for evaluation in prior work [4].
>
> &nbsp;
>
> **Reference**
>
> [1] Chen et al. Do llm evaluators prefer themselves for a reason?
>
> [2] Tan et al. Judgebench: A benchmark for evaluating llm-based judges
>
> [3] Zhang et al. Generative Verifiers: Reward Modeling as Next-Token Prediction
>
> [4] Chen et al. RM-R1: Reward Modeling as Reasoning

---

### Official Review · Reviewer_TAxB · 2025-11-02

**Soundness:** 2
**Presentation:** 2
**Contribution:** 2
**Rating:** 2
**Confidence:** 2

**Summary:**

This work presents an empirical study on the verification dynamics and its implications to TTS. The analysis is centered around problem difficulty, generator and verifier capability. The main findings of this work: verifiers works better on easy problems, weak generators, and further, verification performance is affected by verifier generation capability with a different trend dependent on problem difficulty. The TTS applications inspired by these findings focus on matching the gap between weak and strong generators, and verifiers respectively, which provides guidance of effective TTS design.

**Strengths:**

The results are comprehensive, including the wide range of evaluated models (instruct models and reasoning models as discussed in appendix) and discussed aspects from problem difficulty to generator/verifier capability.

**Weaknesses:**

a). one main theme of the analysis is problem difficulty, but in reality, we do not know the difficulty labels beforehand, then how do we know if a question is on the difficulty extreme where verifier engineering is shown to be less effective? so its practicality on TTS applications is a bit limited in my opinion.

b). efficiency analysis is lacking: is there a compute-optimal TTS approach given a task and a list of candidate generators/verifiers. i imagine something like [1] can be very helpful here.

c). authors should improve task accuracy analysis, besides the verification performances, especially in the TTS applications part. so that the gap between oracle accuracy and verified accuracy can be compared and see how promising each combination is. (is this reported as pass rate in Figure 5a? not entirely follow the metric in the figure) Further, how to choose the best verifier/generator configuration given a task?

d). last but not the least, this work focus on generative verifiers from a LLM-as-a-judge perspective, without fine-tuning; but prior work [2] mentioned that vanilla LLM doesn't perform well on verification. i think the authors should compare the effectiveness of the proposed approach (or the converged optimal selection as discussed above) with the SOTA verifiers beyond LLM-judge, e.g. the top ones on rewardbench.

[1] https://arxiv.org/abs/2502.06703
[2] https://arxiv.org/abs/2408.15240

**Questions:**

Q1. in Figure 2, are verifiers and generators the same? somehow i missed this information.

Q2. there seems a typo in the caption of Figure 5: why are (a-c) and (d-f) mentioned?

Q3. in the final paragraph, line 482, "as weaker generators can approach the post-verification performance of stronger ones when paired with a fixed verifier.", isn't there a specific condition, e.g. decently strong fixed verifier like gpt-4o, otherwise the weak verifier cannot close the gap between weak and strong generators?

Q4. in the paragraph of line 371, "Verifier generation capability influences verification accuracy differently based on problem difficulty", how does the difficulty defined here? my understanding is that the questions are split based on the generation capability (i.e. x-axis), but why are there overlaps between the three difficulty groups?

Q5. Figure 9 (d-f) shows a different trend of reasoning LLMs from instruct models, does the main discussion in RQ1 still hold?

Q6. most evaluated models are less than 100B, why? do authors think the same trend will still hold for larger models such as deepseek V3 or Qwen-235B?

---

> ### Author Response · Authors · 2025-11-21
> **Response to Reviewer TAxB (Part 1)**
>
> > Weakness a: one main theme of the analysis is problem difficulty, but in reality, we do not know the difficulty labels beforehand, then how do we know if a question is on the difficulty extreme where verifier engineering is shown to be less effective? so its practicality on TTS applications is a bit limited in my opinion.
>
> **Prior work has shown that problem difficulty can be estimated for improving TTS.** Recent studies have proposed several practical approaches for estimating question difficulty, including uncertainty-based [2], prompt-based [4], and fine-tuning [3,5] methods. Some of these methods have been applied in the TTS method [2,4,5] and demonstrated improvements.
>
> **Our findings hold even when difficulty is estimated using label-free estimators; See Appendix D.1, Figures 22 and 23.** We conduct a set of new experiments showing that our findings for RQ1 and RQ3, which depend on oracle measures of problem difficulty, hold even under imperfect measures of problem difficulty that can be computed at test-time. We adopt the uncertainty-based estimator of [2]. As shown in Figures 22 and 23, there are minimal changes in the findings to RQ1 and RQ3, showing that our findings can be applied using "practical" problem difficulty estimates.
>
> Concretely, for each question, we draw 8 completions from Qwen2.5-3B-instruct, extract the final answers, compute an empirical answer distribution, from which we compute a normalized entropy in [0, 1]. A score close to 1 indicates that all 8 answers differ, corresponding to high model uncertainty and therefore high estimated difficulty; full implementation details are reported in Appendix D.1.
>
> Finally, we emphasize that **difficulty estimation is not the sole focus of our analysis work.** Our goal is to analyze the fundamental factors that influence verification effectiveness. To this end, we follow prior analysis-oriented studies on problem difficulty [1] and use oracle difficulty, computed using extensive model sampling and ground-truth labels. This allows us to measure the correlation between difficulty and verification performance without introducing estimation error. The purpose of this section is analytical rather than algorithmic, and the use of oracle difficulty is standard in such settings.
>
> **Reference**
>
> [1] Snell et al. Scaling LLM Test-Time Compute Optimally can be More Effective than Scaling Model Parameters
>
> [2] Lee et al. Semantic exploration with adaptive gating for efficient problem solving with language models
>
> [3] Zhu et al. The LLM Already Knows: Estimating LLM-Perceived Question Difficulty via Hidden Representations
>
> [4] Wang et al. Make every penny count: Difficulty-adaptive self-consistency for cost-efficient reasoning
>
> [5] Damani et al. Learning How Hard to Think: Input-Adaptive Allocation of LM Computation

---

> ### Author Response · Authors · 2025-11-21
> **Response to Reviewer TAxB (Part 2)**
>
> > Weakness b: efficiency analysis is lacking: is there a compute-optimal TTS approach given a task and a list of candidate generators/verifiers. i imagine something like [1] can be very helpful here.
>
> > Weakness c (part 2): Further, how to choose the best verifier/generator configuration given a task?
>
> We incorporate difficulty-stratified analysis and efficiency metrics used in [1] for a new study of selecting models in TTS. We decompose the reviewer's question into two subproblems and address each with targeted experiments. This study is updated in Appendix D.2.
>
> **RQ4.1: How to choose generators given a fixed verifier?** We compare three generator models (Gemma2-2B, 9B, and 27B) under a fixed verifier GPT-4o and analyze how the preferred generator changes across three difficulty ranges (easy, medium, hard). Concretely, we assume access to a *validation set* on which we can compute verification metrics (e.g., balanced accuracy), and show that this validation set allows us to reliably identify the compute optimal generator for an unseen test set. Our main results are presented in Figures 24-27 in Appendix D.2.1. Here, we discuss some high-level trends:
> - On easier problems, validation set verification performance shows that small model + verifier approaches larger models + verifier. This informs practitioners that using a smaller model is computationally friendly while retaining system performance. This performance transfers to the test set performance.
> - On hard problems, we observe the opposite: validation set performance indicates that larger models bring non-negligible performance gains over smaller generators. Again, this holds on the test set.
>
> Overall, we find that our proposed metrics, coupled with a validation set, allow for stable prediction of the most efficient way to use inference-time compute. To measure efficiency, we compute the combined inference FLOPS per problem between the generator and verifier. To measure overall performance, we report task accuracies computed via pass rate (i.e., the accuracy in expectation) and majority vote. Complete implementation details are presented in Appendix D.2.1.
>
> **RQ5.1: How to choose verifiers given a fixed generator?** We compare two verifiers (GPT-4o and Qwen2.5-7B) using a fixed generator. The efficiency and accuracy metrics follow those used in RQ4.1. Details are given in Appendix D.2.2 and results are in Figure 28-29.
>
> This setup mirrors that of RQ4.1 above: We assume access to a validation set, and use validation set performance to select the optimal verifier. Concretely, if we deem a larger verifier does not verify substantially better than a smaller one, we simply choose the smaller verifier for TTS to save compute. On the hardest problems, GPT-4o provides only a small improvement in verification accuracy over Qwen2.5-7B, and the test set results confirm that the stronger verifier yields only marginal TTS gains despite higher compute cost. On hard problems, GPT-4o shows a clear verification-accuracy advantage, and selecting it leads to substantially higher TTS accuracy on the test set. Overall, verification accuracy provides a reliable signal: when the metric gap between the two verifiers is small, the smaller verifier is the more cost-efficient choice; when the gap is large, the stronger verifier yields meaningful additional improvements.
>
> &nbsp;
>
> > Weakness b: efficiency analysis is lacking: is there a compute-optimal TTS approach given a task and a list of candidate generators/verifiers. i imagine something like [1] can be very helpful here.
>
> We clarify how our work differs from proposing a compute-optimal approach in [1]. Here are the differences:
> * [1] study which TTS method (Best-of-N, beam search, lookahead search) performs best under different configurations of problem difficulty, PRM, and policy model. Their goal is to select the best TTS algorithm for each configuration, and the combined selection is termed compute-optimal. Our TTS analysis, on the other hand, focuses on how the verification dynamics identified in RQ1–RQ3 Section 4 (problem difficulty, generator capability, verifier capability) translate into concrete differences in TTS performance.
> * [1] identify that one TTS method choice can outperform another under different scenarios. We do not observe cases where a weaker generator/verifier choice outperforms a stronger one under our TTS setup. Instead, we find that the TTS performance gap between small and large models can be reduced. This depends on verification effectiveness, which differs based on model pairs and problem difficulty. In these situations, we can reduce compute by selecting a weaker generator/verifier while preserving a large fraction of gains.
>
> &nbsp;
>
> **Reference**
>
> [1] Liu et al. Can 1B LLM Surpass 405B LLM? Rethinking Compute-Optimal Test-Time Scaling

---

> ### Author Response · Authors · 2025-11-21
> **Response to Reviewer TAxB (Part 3)**
>
> > Weakness c (part 1): authors should improve task accuracy analysis, besides the verification performances, especially in the TTS applications part. so that the gap between oracle accuracy and verified accuracy can be compared and see how promising each combination is. (is this reported as pass rate in Figure 5a? not entirely follow the metric in the figure)
>
> **Clarifying TTS accuracy metric and analysis** As defined in Section 3.1 (L188), we use the pass rates to measure TTS performance without and with verification. As noted on L202, these metrics represent the expected accuracy obtained by uniformly sampling one response from the model-generated samples or from the verifier-retained samples. These metrics are meant to demonstrate the *expected* performance using a verifier, decoupling TTS performance from any particular response selection strategy.
>
> Our TTS analysis in Section 5 hinges on analyzing gains in expectation. We report the pass rates with and without verification, as well as their difference (verification gain). This analysis is meant to concretely demonstrate that our results in RQ1-RQ3 manifest in tangible and predictable ways in practical settings, not just in controlled, static evaluation settings. We complement the pass rate metric with a concrete selection strategy, majority vote, included in the new experiments introduced above (Appendix D.2).
>
> &nbsp;
>
> > Weakness d). last but not the least, this work focus on generative verifiers from a LLM-as-a-judge perspective, without fine-tuning; but prior work [2] mentioned that vanilla LLM doesn't perform well on verification. i think the authors should compare the effectiveness of the proposed approach (or the converged optimal selection as discussed above) with the SOTA verifiers beyond LLM-judge, e.g. the top ones on rewardbench.
>
> We wish to clarify and emphasize that we intentionally exclude finetuned verifiers since **our main research aim is to analyze the factors that influence verification dynamics**. Likewise, *we are not proposing a new benchmark or methodology for which we compare general-purpose models with specialized verifiers*.
>
> We performed initial evaluations with finetuned verifiers that revealed several shortcomings in finetuned evaluators for analysis purposes:
> - Scaler reward models are incapable of generation, which does not allow us to estimate metrics like the generator capability of problem difficulty. Moreover, such evaluators require extensive manual tuning of score cut-off thresholds in reference-free settings that may not necessarily transfer from domain to domain.
> - Finetuned generative reward models are (1) often pairwise evaluators (i.e., require *reference answers*) and (2) often overfit to the specific task, failing to output coherent outputs for non-verification tasks. As a concrete example towards (2), Skywork-Critic-8B cannot output responses beyond [A] or [B], indicating extreme overfitting. Again, we cannot use such models to estimate the generator capability of problem difficulty.
>
> As a result, our analysis in RQ1-RQ3 focuses on general-purpose models to most accurately measure things like generator capability and problem difficulty. Our TTS experiments were to demonstrate that the findings in RQ1-RQ3 manifest not just in controlled settings but also predictably influence performance in practical settings.
>
> We do agree that analyzing finetuned verifiers is a separate, important question. We believe our work can inform the development best-practices of future verification benchmarks to provide the most "signal". As a concrete example, we found that verifiers tended to provide very little benefit in TTS scaling settings when (1) the problems are extremely difficult and (2) the generator is very strong. We believe future benchmarks should focus on this setting, generating responses to hard problems from strong generators. Here, an "ideal" verifier should be able to help regardless of problem difficulty and generator capability, and meaningful progress in this regime is a clear signal of improvement.
>
> &nbsp;
>
> > Q1. in Figure 2, are verifiers and generators the same? somehow i missed this information.
>
> The verification evaluation setup is described in the paragraph “Verification Metrics and Evaluation” in Section 3.1. We use the same set of 15 general-purpose instruction models as both generators and verifiers, using different prompts. For each problem, every generator samples 64 responses, and we then draw 8 balanced correct and incorrect responses for evaluating verification. In Figure 2, each verifier evaluates 8 responses per generator across all 15 generator models, giving 120 responses per problem.
>
> &nbsp;
>
> > Q2. there seems a typo in the caption of Figure 5: why are (a-c) and (d-f) mentioned?
>
> We thank the reviewer for pointing out the typo. We have fixed them to be (a) and (b) in the updated draft.

---

> ### Author Response · Authors · 2025-11-21
> **Response to Reviewer TAxB (Part 4)**
>
> > Q3. in the final paragraph, line 482, "as weaker generators can approach the post-verification performance of stronger ones when paired with a fixed verifier.", isn't there a specific condition, e.g. decently strong fixed verifier like gpt-4o, otherwise the weak verifier cannot close the gap between weak and strong generators?
>
> We find that whether the gap can be closed is correlated with problem difficulty and verification accuracy. This correlation is consistent across all of our experiments.
>
> For hard problems, verification does not close the performance gap between weaker and stronger generators. This is shown in Appendix C.5 (Figure 16, second row, difficulty ranges $[0.2, 0.4)$). In these ranges, the performance gap between smaller and larger generators (e.g., Gemma2-9B vs. Gemma2-27B) remains unchanged after verification. The verification metrics (green and purple curves) in the same row explain this: the TPR of the weaker generators is significantly lower, and the stronger generator obtain higher verification accuracy (average of TPR and TNR). Therefore, the stronger generator receives better verification and higher verification gain, so its TTS performance advantage persists.
>
> In the medium and easy difficulty ranges (Figure 16, rows 3–5, difficulty $[0.4,1.0)$), the smaller or medium-sized generators obtain higher verification accuracy. These are the ranges where we observed that weaker generators can approach the post-verification performance of stronger ones in the main paper. In these settings, verification naturally reduces the performance gap, since the weaker generators receive higher verification accuracy and therefore reach comparable TTS performance as the larger models.
>
> In summary, verification closes the generator gap when the weaker generator obtains higher verification accuracy than the larger one. When the stronger generator achieves higher verification accuracy than the weaker one, the gap remains.
>
> &nbsp;
>
> > Q4. in the paragraph of line 371, "Verifier generation capability influences verification accuracy differently based on problem difficulty", how does the difficulty defined here? my understanding is that the questions are split based on the generation capability (i.e. x-axis), but why are there overlaps between the three difficulty groups?
>
> We first clarify how difficulty and generation capability are defined, since Figure 6 uses two different quantities. As stated in L171, the difficulty of each problem is defined using the average pass rate across all 15 generator models. A problem is labeled "easy", "medium", or "hard" based on this average value. For example, in Figure 6, problems with average pass rate in $[0.8, 0.9)$ are assigned to the easy group. The x-axis in the figure shows each verifier’s own generation capability, i.e., its individual pass rate on that same set of problems. This value is model-specific and is not the same as the difficulty score.
>
> Therefore, even within an "easy" difficulty group, not all models will achieve high pass rates. Most models perform well on easy problems, but some models may still fail on a subset of them. Their individual pass rate can therefore be low (e.g., around 0.2) even though the average pass rate across all models is high. This naturally creates overlapping x-axis values across difficulty groups. Such overlap is expected and simply reflects variation in model generation capability.
>
> &nbsp;
>
> > Q5. Figure 9 (d-f) shows a different trend of reasoning LLMs from instruct models, does the main discussion in RQ1 still hold?
>
> We do not believe there is a fundamental mismatch between results presented in Figure 11(d-f) (before revision, Figure 9) and those presented in Figure 2, as *our findings for RQ1,  which form the basis of Figure 2, do not rely at all on TNR*: We draw all conclusions from TPR, for which reasoning models and instruction tuned models exhibit the same TPR behavior as a function of problem difficulty. Regarding TNR vs. problem difficulty, Figure 2 shows that for some instruction-tuned models, TNR increases vs. problem difficulty, whereas for others it decreases; Overall, the impact of problem difficulty on TNR is extremely mixed, varying on a model-to-model basis. Moreover, our results for TNR, the focus on RQ2, also hold with reasoning models, as shown in Figure 13 and supporting discussion in Appendix C.2.

---

> ### Author Response · Authors · 2025-11-21
> **Response to Reviewer TAxB (Part 5)**
>
> > Q6. most evaluated models are less than 100B, why? do authors think the same trend will still hold for larger models such as deepseek V3 or Qwen-235B?
>
> We included additional experiments on a larger model, Qwen3-235B-A22B-Instruct-2507, with details updated in Appendix D.3 and results shown in Figures 30. These results confirm that the main trends identified in RQ1 and RQ2 continue to hold at this larger scale.
>
> We also note that our experiments included GPT-4o. Its exact size is not public, but it is generally believed to operate at a scale above 100B active parameters. Therefore, our evaluation already covers models in this range, and the new Qwen-235B results further support the consistency of the trends.

---

### Official Review · Reviewer_PDQk · 2025-11-03

**Soundness:** 3
**Presentation:** 3
**Contribution:** 3
**Rating:** 6
**Confidence:** 3

**Summary:**

The paper presents an experimental study of generative verifiers across 15 models of diverse sizes on 12 benchmarks.
In their extensive evaluation, they make the following observations:
1. Problem difficulty has an inverse correlation with the True Positive Rate (TPR) of the verifier, i.e., harder problems have a lower TPR of the verifier, while this is not the case for the True Negative Rate (TNR).
2. The capability of the LLM in answer generation correlates positively with the error detection capability in verification, and errors made by stronger LLMs are harder to detect than those made by weaker LLMs.
3. Verifier capability is positively correlated with verification performance, but it is non-linear for very easy and very hard problems.

The paper further makes the following observations when applying these findings to Test-Time scaling (TTS):

4. A weak generator can benefit much more from a strong verifier.

5. Test time scaling helps especially for medium difficulty problems, and less or not for very easy or very hard problems.

**Strengths:**

- overall solid experimental evaluation and setup (some limitations see below).
- Overall very extensive study w.r.t. datasets and models.
- A very well-structured formulation of 5 research questions and answering them experimentally.

**Weaknesses:**

## Major
1. Some of the findings are maybe unsurprising or it is difficult to benefit from the the findings in practice. E.g. Using a very strong verifier (GPT 4o) on weaker answer models seems of limited practical use.
Or that verifiers do not provide much benefit for very easy or very difficult problems: while interesting, the paper does not examine how to reliably identify very hard and very difficult problems, but is able to group by difficulty based on ground truth, in practice one is likely confronted with easier and more difficult problems without knowing which ones are easier and which ones are more difficult.
2. RQ4: Is it relevant to verify a weak model with a much stronger verifier? Given the cost of verification (see e.g. Singhi et al., 25), would GPT 4o (w/o verification) already be as good or better? Or just use multiple GPT 4o answerers.
2. Related to the previous point, I miss the point GPT 4o in the x-axis in Figures 5 (a), (b).


## Minor
4. Most References are given as arXiv versions. If they are published, the published version should be cited.

**Questions:**

1. Plot 1(b) and corresponding observation: Is the hidden variable in this plot not the problems that a generator cannot solve, and in this sense, related to the problem difficulty?

---

> ### Author Response · Authors · 2025-11-22
> **Response to Reviewer PDQk (Part 1)**
>
> We thank the reviewer for their review. We're pleased you found our experimental setup "solid", "extensive", and "well-structured". We respond to comments below.
>
> &nbsp;
>
> > Part 1 of Weakness 1. Some of the findings are maybe unsurprising or it is difficult to benefit from the the findings in practice. E.g. Using a very strong verifier (GPT 4o) on weaker answer models seems of limited practical use.
>
> > Weakness 2. RQ4: Is it relevant to verify a weak model with a much stronger verifier? Given the cost of verification (see e.g. Singhi et al., 25), would GPT 4o (w/o verification) already be as good or better? Or just use multiple GPT 4o answerers.
>
> On its face, using a larger verifier for a smaller generator can seem counterintuitive. In Appendix D.5, we present a case study that investigates two setups: Using just GPT-4o as the generator and using GPT-4o as the verifier paired with Qwen3-4B as the generator, where we assume that GPT-4o represents a minimum viable level of performance. We find the following: Both setups achieve the same performance, but using GPT-4o as a verifier consumes 193 tokens per prompt on average, whereas GPT-4o as a generator uses 483 tokens per prompt on average. This represents a 2.5$\times$ reduction in token usage (and thus cost) per prompt. Thus, for practitioners who host small models but care about maximizing performance per dollar, leveraging the relatively short-form nature of verification with API models represents massive savings.
>
> &nbsp;
>
> > Part 2 of Weakness 1. Or that verifiers do not provide much benefit for very easy or very difficult problems:
>
> Our complete finding for RQ5 shows that the gap between small and large verifiers varies with problem difficulty and is correlated with verification metrics. In Appendix D.2.2 (Figure 28 and Figure 29), we use our findings to select the best-sized verifier under compute constraints. Specifically, we show that, if given access to a validation set where we can compute metrics like balanced accuracy, we are able to determine the compute optimal verifier for an unseen test set. That is, at test-time, if we can estimate problem difficulty, then we can select a verifier that brings compute-informed benefits: If we deem a problem to be very easy or hard, then we simply do not need to expend compute to verify the problem.
>
> It is reasonable to ask if it is feasible to determine problem correctness in lieu of ground-truth labels at test-time. Our results using entropy-based estimators of problem difficulty proved to be very reliable; We discuss this next.
>
> &nbsp;
>
> > Part 3 of Weakness 1. while interesting, the paper does not examine how to reliably identify very hard and very difficult problems, but is able to group by difficulty based on ground truth, in practice one is likely confronted with easier and more difficult problems without knowing which ones are easier and which ones are more difficult.
>
> This is a very practical concern. We chose an "oracle" measure of problem difficulty to enable fundamental analysis free of difficulty measurement error; Our setup follows [1]. However, recent work has shown that problem difficulty can be estimated for improving TTS, including uncertainty-based [2], prompt-based [4], and fine-tuning [3,5] methods. Some of these methods have been applied in the TTS method [2,4,5] and demonstrated improvements.
>
> Towards this end, we experiment with the uncertainty-based estimator of [2], and show that when using problem difficulty estimates from a relatively weak model, Qwen2.5-3B-Instruct, we are able to closely replicate our findings in Figures 2 and 6, which were based on oracle difficulty measures. This shows that our main conclusions about verification dynamics continue to hold even under an approximate, uncertainty-based difficulty estimator.
>
> Full results and implementation details are presented in Appendix D.1 and Figures 22 and 23. Concretely, for each question, we draw 8 completions from Qwen2.5-3B-instruct, extract the final answers, compute the empirical answer distribution, and compute the normalized entropy in [0, 1]. A score close to 1 indicates that all 8 answers differ, corresponding to high model uncertainty and, therefore, high estimated difficulty.
>
> &nbsp;
>
> > Weakness 3. Related to the previous point, I miss the point GPT 4o in the x-axis in Figures 5 (a), (b).
>
> For Weakness 3, we clarify that in Figure 7 (a,b), the GPT-4o is written in lowercase (gpt-4o), and is the fourth model from the right.
>
> &nbsp;
>
> > Weakness 4. Most References are given as arXiv versions. If they are published, the published version should be cited.
>
> We thank the reviewer for pointing this out. We have updated the references to cite the published versions in the revised draft.

---

> ### Author Response · Authors · 2025-11-22
> **Response to Reviewer PDQk (Part 2)**
>
> > Question 1 Plot 1(b) and corresponding observation: Is the hidden variable in this plot not the problems that a generator cannot solve, and in this sense, related to the problem difficulty?
>
> We first provide our interpretation of the reviewer’s question. In Figure 1(b), there are three plots corresponding to RQ1–RQ3. The RQ1 plot (left) and the RQ3 plot (right) are directly related to problem difficulty, since RQ1 uses difficulty as an independent variable and RQ3 conditions on difficulty. The middle plot corresponds to RQ2, which analyzes how generator capability affects the true negative rate (TNR), that is, the verifier’s ability to catch generator errors. We interpret the reviewer’s question as asking whether problem difficulty is a hidden variable in this RQ2 plot.
>
> Under this interpretation, we clarify that our design explicitly reduces the influence of problem difficulty in the RQ2 analysis. As stated in L317, for TNR, we use only problems for which all generators produce at least one wrong response. This removes problems that are too easy for some generators and ensures that the problem set used for computing TNR is the same across generators. By taking this design, we reduce the confounding effect of problem difficulty.
>
> If this is not the reviewer’s intended question, we are happy to clarify further after receiving additional feedback.
>
> &nbsp;
>
> **Reference**
>
> [1] Snell et al. Scaling LLM Test-Time Compute Optimally can be More Effective than Scaling Model Parameters
>
> [2] Lee et al. Semantic exploration with adaptive gating for efficient problem solving with language models.
>
> [3] Zhu et al. The LLM Already Knows: Estimating LLM-Perceived Question Difficulty via Hidden Representations
>
> [4] Wang et al. Make every penny count: Difficulty-adaptive self-consistency for cost-efficient reasoning
>
> [5] Damani et al. Learning How Hard to Think: Input-Adaptive Allocation of LM Computation

---

### Author Response · Authors · 2025-11-22
**Authors’ Global Response and Summary of Revisions**

Thank you to all the reviewers for taking the time to read and review our work. We are glad that reviewers found our work to be "timely", "comprehensive", and "well-structured”. We appreciate all the comments and questions, which help us improve our paper. We have uploaded a revised draft with all changes marked in red. All references to line numbers, sections, and figures in the rebuttal correspond to the revised draft. Below, we summarize the major updates.

- **Difficulty estimation.** We evaluate our main findings using a practical, label-free difficulty estimator and show that all key trends remain consistent.  (Appendix D.1)
- **Application of the verification findings to TTS efficiency.** We add an extended study showing how the identified verification dynamics can guide generator and verifier choices in TTS, together with improved efficiency analysis.  (Appendix D.2)
- **Interpretability analysis.** We strengthen the explanations behind RQ1 and RQ2 with systematic large-scale analyses on more than 400K verification and generation traces.  (L281 Figure 3, L356 Figure 5)
- **Robustness of the main findings.** We include ablations using a >100B model (Qwen3-235B) and additional verification prompts, confirming that the verification dynamics hold across models and prompting strategies.  (Appendix D.3, D.4)
- **Practicality of using strong verifiers.** We add an efficiency study comparing strong models used as generators vs. verifiers, showing that verification can be more computationally efficient in realistic settings.  (Appendix D.5)

We hope this addresses the main concerns of the reviewers. Please see below for our responses to each comment.

---

### Author Response · Authors · 2025-11-30
**Authors’ Summary of Rebuttal**

Dear Area Chair,

To support the decision-making process, we summarize the current status of our submission below. We have made substantial efforts to address the reviewers' comments, and a summary of revisions is provided in the public comment **Authors’ Global Response and Summary of Revisions**. We believe we have addressed all concerns raised. In particular, Reviewer 1wQz (score: 8) provided a detailed and thorough review that already touched on many of the concerns raised by the other reviewers. Our responses to Reviewer 1wQz incorporate all major additions outlined in our global summary.

We are glad that Reviewer 1wQz read through our rebuttal and confirmed that they are maintaining their original score of 8:

> Thank you for your comprehensive response! After reading the rebuttal, I will maintain my positive score.

Reviewers TAxB, QrzN, and PDQk did not provide responses before the discussion was prematurely closed.

Thank you for handling our submission.

Yours sincerely,

#15980 Authors

---

### Meta-Review · Area_Chair_sjWF · 2026-01-08

**Summary:**

The reviewers acknowledged the comprehensive nature of the study (12 benchmarks, 15 models) and the timely relevance of investigating verification dynamics for test-time scaling. The primary concerns centered on the practicality (e.g., relying on difficulty labels, the cost-effectiveness of strong verifiers on weak generators) and the lack of mechanistic explanations for the observed "intuitive" patterns. The rebuttal addressed these concerns.

Therefore, AC recommends acceptance at this stage.

**Reviewer Concerns:**

Addressed:
1. Difficulty estimation: Reviewers questioned whether the findings depended on difficulty labels, which are unavailable at test time. The authors addressed this by replicating their main findings using a practical, uncertainty-based difficulty estimator.
2. Deep Insights: Reviewers argued the findings (e.g., "easy problems are easier to verify") were intuitive but lacked depth. The rebuttal added quantitative analysis showing that verification failures on hard problems are driven by the verifier's inability to generate correct internal references, and detection of weak generators is driven by surface-level inconsistencies.
3. Cost/Efficiency of Strong Verifiers: Reviewer questioned the economic sense of using a strong verifier (GPT-4o) on a weak generator. The authors provided a new case study (Appendix D.5) showing that Weak Generator + Strong Verifier can match the performance of Strong Generator alone while reducing token consumption by ~2.5x.
4. Robustness: Reviewers asked for larger models and prompt robustness. The authors added experiments with Qwen3-235B and alternative prompting strategies.

Still Outstanding:
No outstanding concerns.

**Reviewer Scores:**

- Reviewer 1wQz: 8.
- Reviewer PDQk: remains 6.
- Reviewer QrzN: remains 4. The authors provided mechanistic explanations about reference generation and surface errors. I think the reviewer could raise the score to 6.
- Reviewer TAXB: remains 2. The reviewer questioned the efficiency and practicality. The rebuttal addressed the difficulty estimation and efficiency/model selection. I think the reviewer could raise the score to 4/6.

---

### Decision · Program_Chairs · 2026-01-26

Accept (Poster)